# Is an artificial limb embodied as a hand? Brain decoding in prosthetic limb users

Roni O. Maimon-Mor [1,2], Tamar R. Makin [1,2,3]*

1 Institute of Cognitive Neuroscience, University College London, London, United Kingdom, 2 WIN Centre, Nuffield Department of Clinical Neuroscience, University of Oxford, Oxford, United Kingdom, 3 Wellcome Centre for Human Neuroimaging, University College London, London, United Kingdom

* t.makin@ucl.ac.uk

## Abstract

The potential ability of the human brain to represent an artificial limb as a body part (embodiment) has been inspiring engineers, clinicians, and scientists as a means to optimise human–machine interfaces. Using functional MRI (fMRI), we studied whether neural embodiment actually occurs in prosthesis users' occipitotemporal cortex (OTC). Compared with controls, different prostheses types were visually represented more similarly to each other, relative to hands and tools, indicating the emergence of a dissociated prosthesis categorisation. Greater daily life prosthesis usage correlated positively with greater prosthesis categorisation. Moreover, when comparing prosthesis users' representation of their own prosthesis to controls' representation of a similar looking prosthesis, prosthesis users represented their own prosthesis more dissimilarly to hands, challenging current views of visual prosthesis embodiment. Our results reveal a use-dependent neural correlate for wearable technology adoption, demonstrating adaptive use–related plasticity within the OTC. Because these neural correlates were independent of the prostheses' appearance and control, our findings offer new opportunities for prosthesis design by lifting restrictions imposed by the embodiment theory for artificial limbs.

**Data Availability Statement:** Full study protocol, key materials and full clinical/demographic details are available from the Open Science Framework at https://osf.io/kd2yh/ Data used in all the reported analyses and figures are available from the Open

## Introduction

The development of wearable technology for substitution (e.g., prosthetic limbs [1], exoskeletons [2]) and augmentation (e.g., supernumerary fingers and arms [3]) is rapidly advancing. Clinical research on prosthetic limbs, the most established form of wearable motor technology to date, teaches us that technological development is necessary but not sufficient for successful device adoption and usage. For example, only 45% of all arm amputees choose to use their prosthesis regularly [4]. The causes for prosthesis rejection are multiplex and include awkward control over the device, lack of tactile feedback, and complex training requirements [4–6]. Crucially, successful prostheses adoption depends on the human brain's ability to effectively represent and operate it [7]. A popular (and yet untested) assumption is that amputees reject their prostheses—tools designed to substitute hand function—because they do not feel like it is

Science Framework at https://osf.io/4mw2t/ Full raw data including each participant/ROI fMRI BOLD activity values across voxels/conditions/runs, will be available upon request.

**Funding:** This work was supported by a Wellcome Trust Senior Research Fellowship (https://wellcome.ac.uk/, grant number: 215575/Z/19/Z), and an ERC Starting Grant (https://erc.europa.eu/, grant number: 715022 EmbodiedTech) and a Sir Henry Dale Fellowship jointly funded by the Wellcome Trust (https://wellcome.ac.uk/) and the Royal Society (https://royalsociety.org/, 104128/Z/14/Z), awarded to T.R.M. R.O.M.M. is supported by the Clarendon scholarship (http://www.ox.ac.uk/clarendon) and University College, Oxford (http://www.univ.ox.ac.uk/). The funders had no role in study design, data collection and analysis, decision to publish, or preparation of the manuscript.

**Competing interests:** The authors have declared that no competing interests exist

**Abbreviations:** ANCOVA, analysis of covariance; EBA, extrastriate body-selective area; fMRI, functional MRI; FSL, FMRIB's Software Library; hIP, human intraparietal; IPS, intraparietal sulcus; OTC, occipitotemporal cortex; PAL, prosthesis activity log; ROI, region of interest.

a real body part (i.e., embodied) [8] and that embodiment can improve prosthesis usage [9–16].

The first challenge in harnessing the potential powers of embodiment to improve prosthesis design and usage is measuring embodiment. Embodiment is an umbrella term for the multisensory association of external objects with body representation, which engages multiple levels from both conceptual and perceptual perspectives (see "Discussion"). From a neural standpoint, embodiment is defined by the successful allocation of brain resources, originally devoted to controlling one's own body, to represent and operate external objects [17]. Here, we focus on visual embodiment—the successful allocation of visual hand-related neural resources. Visual embodiment is particularly relevant for studying prosthesis representation; this is because prosthesis usage is highly visually guided [18]. Moreover, visual internal models of the body have been suggested as essential gateways for processing multisensory integration that will result in successful bodily ownership [19], a desired feature of prosthesis usage [9,10]. Previous efforts to test visual prosthesis embodiment have been centred on an illusion of body ownership over the prosthesis (most commonly, the rubber hand illusion, which relies on visual manipulations) with mixed success [11,20–23]. Crucially, such efforts focused on measuring visual embodiment but did not associate it with improved prosthesis usage [11,20–26] (though see the work by Graczyk and colleagues [27] for results from 2 individuals). We have recently found that as a whole, one-handed individuals tend to respond neutrally to embodiment statements in relation to their own prostheses [28]. Moreover, these measures of embodiment tend to rely on an explicit sense of body ownership, which might not be a necessary consequence of implicit neuronal embodiment (i.e., reallocation of body part resources to represent or control the prosthesis [17]).

As a more direct means of measuring neural visual embodiment, or how the brain represents prosthetic limbs, we recently assessed activity levels in prosthesis users' occipitotemporal cortex (OTC) while participants were viewing images of prosthetic limbs, using functional MRI (fMRI) [29]. The OTC is known to contain distinct visual representations of different object categories [30], e.g., hands [31,32], tools [33], faces [34], and bodies [35]. It was previously shown to contain overlapping visual and motor body part selective representations [31,36] and even respond to touch [37,38]. OTC has been previously implicated in multisensory integration processing related to embodiment [39–41] and OTC's connectome associates it with hand actions. For example, hand- and body-selective visual areas uniquely connect to the primary sensorimotor hand area [37]. These characteristics qualify the OTC as an ideal candidate for studying action-related visual body representation. We previously found that individuals who used their prosthesis more in daily lives also showed greater activity in OTC's hand-selective visual areas when viewing images of prostheses and greater functional coupling with sensorimotor hand areas [29]. This result demonstrates that prosthesis users are able to engage visuomotor hand-selective areas while passively representing a prosthesis. However, it is yet unknown whether prosthesis visual representation actually mimics that of a hand (i.e., neural embodiment). Alternatively, because the visual hand-selective regions partially overlap with handheld tool representation [42], the observed activity gains may reflect the prosthesis being represented as a tool. As a third option, because object categorisation in OTC is thought to be based on their semantic and functional properties [43], expert prosthesis usage may result in the emergence of prostheses representation as a new 'category', diverging from its existing natural categories (e.g., 'hands', 'tools'). This third alternative is consistent with recent evidence showing that visual expertise can contribute to the shaping of categorical representation in OTC [44] (see 'Discussion').

Brain decoding techniques that take advantage of multivoxel representational patterns allow us to reveal the representational structural underlying fMRI activity, e.g., to dissociate

overlapping hand and tool representations within the lateral OTC [33,45]. Here, we utilised fMRI data of 32 individuals missing a hand, with varying levels of prosthesis usage (hereafter, 'prosthesis users'; see Table 1) and 24 two-handed control participants, who have varying life experience of viewing prostheses (See S1 Table). The OTC's extrastriate body-selective area [35] was independently localised. Two main hypotheses were tested: the embodiment hypothesis, assessing whether prosthetic limbs are in fact represented visually as hands and not tools and the categorisation hypothesis, assessing whether a new 'prosthesis' category has formed. To provide distinct predictions for each of these hypotheses, we studied representational similarities between hand, tools, and upper-limb prosthetics images (both cosmetic—designed to resemble hand appearances—and active—designed to afford a grip, e.g., a 'hook'; Fig 1A) and compared prosthesis users to controls. Broadly speaking, the visual hand embodiment hypothesis predicts that compared to controls, the various prosthesis conditions in prosthesis users will be more similar to hands than to tools (notice that this prediction also allows us to test the inverse prediction—that prostheses are represented more like tools in one-handers). The categorisation hypothesis predicts that, in prosthesis users, the prosthesis conditions will be more similar to each other relative to hands and tools.

## Results

### Clustering of prostheses types in prosthesis users but not in controls

Analysis was focused on the OTC's extrastriate body-selective area (EBA) [46,47]. This region of interest (ROI) was independently localised for each participant by choosing the 250 voxels in each hemisphere showing the strongest preference to images of headless bodies over everyday objects (Fig 1B).

To investigate the underlying representational structure within this region, we first characterised multivoxel activity patterns for each participant and condition (hands, tools, cosmetic prostheses that look like hands, and active prostheses that tend to resemble tools rather than hands, Fig 1A; all exemplars are available at https://osf.io/kd2yh/). Distances between each pair of activity patterns (e.g., hands and cosmetic prostheses) were calculated (noise-normalised cross-validated mahalanobis distances [48]). More similar activity patterns will result in smaller distances, or in other words, the more dissimilar patterns are, the greater their relative distance is. Because these are multidimensional patterns, one way to visualise the structure is using a dendrogram, or linkage tree, which illustrates how the different conditions cluster. Using this method on the average distances for each group, we found qualitative differences in the representational structure between controls and prosthesis users (Fig 2). For control participants, cosmetic prostheses were clustered with hands, and active prostheses were clustered with tools, reflecting their native intercategorical similarities across conditions (see S2 Fig for visual similarity analysis). For prosthesis users, however, the 2 prostheses were clustered together, potentially reflecting a newly formed prosthesis category, with tools and hands being further away from both prosthesis conditions.

### Prosthesis-like (categorical) and not hand-like (embodied) representation of prostheses in prosthesis users

Next, we set out to quantify prosthesis representation using the 2 alternative theoretical frameworks—embodiment versus categorisation. According to the embodiment hypothesis, prosthesis representation should resemble hand representation. This hypothesis predicts that in prosthesis users, each of the 2 prosthesis conditions will be more similar (smaller distance) to hands than to tools, compared to controls (quantified as a hand-similarity index; see

**Table 1. Prosthesis users' demographic details and daily prosthesis usage habits.**

| Subject | Gender | Age | Age at amputation | Level of amputation | Missing hand side | Cause | Usage skill (PAL)[a] | Usage time[b] | | | Prosthesis usage score[c] | Own prosthesis condition[d] | Own condition hand-like[e] |
|---|---|---|---|---|---|---|---|---|---|---|---|---|---|
| | | | | | | | | Cosmetic | Active | | | | |
| | | | | | | | | | Mechanical | Myoelectric | | | |
| 01 | Male | 57 | 20 | Below elbow | Left | Trauma | 0.57 | **5** | 0 | 0 | 2.99 | Cosmetic | 1 |
| 02 | Female | 49 | 0 | Below elbow | Left | Congenital | 0.46 | **4** | 0 | 0 | 1.92 | Cosmetic | 1 |
| 03 | Male | 59 | 40 | Above elbow | Left | Trauma | 0 | 0 | 0 | 0 | −2.42 | N/A | N/A |
| 04 | Female | 52 | 0 | Below elbow | Right | Congenital | 0.15 | **5** | 1 | 0 | 1.00 | Cosmetic | 1 |
| 05 | Male | 58 | 27 | Above elbow | Left | Trauma | 0.09 | **5** | 2 | 0 | 0.72 | Mechanical | 0 |
| 06 | Male | 53 | 28 | Below elbow | Left | Trauma | 0.24 | 3 | **5** | 0 | 1.43 | Mechanical | 0 |
| 07 | Male | 52 | 0 | At wrist | Left | Congenital | 0.04 | 0 | **3** | 0 | −0.60 | N/A | N/A |
| 08 | Male | 41 | 27 | Above elbow | Right | Trauma | 0.09 | **2** | 1 | 0 | −0.91 | Cosmetic | 1 |
| 09 | Male | 48 | 17 | Above elbow | Left | Trauma | 0 | **2** | 2 | 0 | −1.34 | N/A | N/A |
| 10 | Female | 25 | 0 | At wrist | Right | Congenital | 0 | 0 | 0 | 0 | −2.42 | Cosmetic | 1 |
| 11 | Male | 49 | 0 | Above elbow | Left | Congenital | 0.26 | 1 | **4** | 0 | 0.98 | Mechanical | 1 |
| 12 | Male | 37 | 27 | Above elbow | Left | Trauma | 0.28 | 0 | **2** | 0 | −0.01 | N/A | N/A |
| 13 | Female | 46 | 38 | Below elbow | Left | Trauma | 0 | 0 | 0 | 0 | −2.42 | Cosmetic | 1 |
| 14 | Female | 28 | 0 | At wrist | Left | Congenital | 0 | 0 | 0 | 0 | −2.42 | N/A | N/A |
| 15 | Male | 64 | 33 | Below elbow | Right | Trauma | 0.33 | 0 | 2 | **5** | 1.85 | Myoelectric | 1 |
| 16 | Male | 38 | 0 | Below elbow | Left | Congenital | 0.39 | **5** | 0 | 0 | 2.14 | Cosmetic | 1 |
| 17 | Female | 24 | 18 | Below elbow | Right | Trauma | 0 | 0 | 0 | 0 | −2.42 | Cosmetic | 1 |
| 18 | Female | 27 | 0 | Below elbow | Left | Congenital | 0.54 | **5** | 0 | 0 | 2.84 | Cosmetic | 1 |
| 19 | Male | 49 | 37 | Above elbow | Left | Trauma | 0 | **1** | 0 | 0 | −1.88 | Cosmetic | 1 |
| 20 | Male | 60 | 0 | At wrist | Left | Congenital | 0.06 | **2** | 0 | 0 | −1.05 | Cosmetic | 1 |
| 21 | Female | 34 | 0 | Below elbow | Right | Congenital | 0.46 | **5** | 0 | 0 | 2.47 | Cosmetic | 1 |
| 22 | Female | 36 | 0 | Below elbow | Right | Congenital | 0.57 | **5** | 0 | 0 | 2.99 | Cosmetic | 1 |
| 23 | Female | 50 | 45 | Above elbow | Left | Tumor | 0 | 0 | **2** | 0 | −1.34 | Mechanical | 0 |
| 24* | Female | 41 | 0 | Below elbow | Left | Congenital | 0.54 | 0 | 0 | **5** | 2.84 | Myoelectric | 1 |
| 25* | Male | 29 | 24 | Through shoulder | Left | Trauma | 0.09 | 0 | 0 | **2** | −0.91 | Myoeletric | 1 |
| 27* | Male | 25 | 0 | Below elbow | Left | Congenital | 0.59 | 1 | 0 | **5** | 3.08 | Myoelectric | 1 |
| 28* | Male | 34 | 0 | At wrist | Left | Congenital | 0.11 | 0 | 0 | **3** | −0.27 | Myoelectric | 1 |
| 29* | Male | 25 | 18 | At wrist | Left | Trauma | 0 | 0 | **2** | 0 | −1.34 | N/A | N/A |

*(Continued)*

**Table 1.** (Continued)

| Subject | Gender | Age | Age at amputation | Level of amputation | Missing hand side | Cause | Usage skill (PAL)[a] | Usage time[b] | | | Prosthesis usage score[c] | Own prosthesis condition[d] | Own condition hand-like[e] |
|---|---|---|---|---|---|---|---|---|---|---|---|---|---|
| | | | | | | | | Cosmetic | Active | | | | |
| | | | | | | | | | Mechanical | Myoelectric | | | |
| 30 | Male | 38 | 0 | Below elbow | Left | Congenital | 0 | 0 | **2** | 1 | −1.34 | Myoelectric | 1 |
| 31 | Female | 49 | 0 | At wrist | Left | Congenital | 0 | **1** | 0 | 0 | −1.88 | Cosmetic | 1 |
| 32 | Male | 45 | 20 | Below elbow | Right | Trauma | 0.09 | **2** | 0 | 0 | −0.91 | Cosmetic | 1 |
| 33 | Male | 32 | 31 | Above elbow | Left | Trauma | 0 | 0 | **2** | 0 | −1.34 | Mechanical | 0 |

[a]How frequently users incorporate their prosthesis in an inventory of 27 daily activities (e.g., taking money out of wallet, etc.). Scores of 0 = never, 1 = sometimes, 2 = very often. The sum of all items was divided by the highest possible score, such that individuals were rated on a scale ranging from 0 to 1.

[b]Prosthesis wear time for each prosthesis type: 0 = never; 1 = rarely; 2 = occasionally; 3 = daily <4 h; 4 = daily 4–8 h, 5 = daily >8 h. Bold indicates the user's primary prosthesis.

[c]A composite measure of prosthesis wear time and skill.

[d]The prosthesis the participant brought on the day of experiment and was shown in the own prosthesis condition.

[e]Relates to the visual appearance of the prosthesis they viewed in the own prosthesis condition. Note that participants 11's prosthesis was a mechanical hook with a glove and is therefore hand-like.

*These participants were presented with a myoelectric prosthesis for the active prosthesis condition.

**Abbreviations:** N/A, not applicable; PAL, prosthesis activity log

"Methods"). Notice that the hand-similarity index is the inverse of a tool-similarity index. Therefore, significantly negative embodiment should be taken as evidence for association of the prosthesis with a tool. When comparing hand-similarity indices based on the multidimensional distances across participant groups, we found no significant differences ($t(54) = 0.47$, $p = 0.64$; Fig 3A). A Bayesian $t$ test provided substantial evidence in support of the null hypothesis (BF = 0.2), i.e., that on average amputees do not visually represent an unfamiliar prosthesis more similarly to a hand than controls. Similar results were observed across the 2 hemispheres (right OTC: $t(54) = 0.46$, $p = 0.65$, left OTC: $t(54) = 0.54$, $p = 0.60$).

It is possible that the effects of embodiment are present in a subset of users that rely on their prosthesis for daily function most but that this effect is masked in the group effect by the wide range of usage in our group of prosthesis users. We therefore further tested the relationship between visual embodiment and prosthesis usage by correlating the hand-similarity index with a prosthesis usage score, reflecting usage time and habits (see 'Methods'). According to the embodiment hypothesis, hand-like prosthesis representation should scale with prosthesis usage. We found no significant correlation between the hand-similarity index and everyday prosthesis usage (or with tool similarity; Pearson's $r(30) = −0.03$, $p = 0.86$), suggesting that prosthesis embodiment is not a strong predictor of usage (Fig 3B).

According to the categorisation hypothesis, prosthesis usage should promote a more independent representation of prostheses. This hypothesis predicts that within our multidimensional space, both prosthesis conditions would move away from the existing natural categories (e.g., the distance between cosmetic prostheses and hands will become larger whereas their distance from tools will become smaller; see Fig 3D) and towards one another (smaller distances between cosmetic and active prostheses). In other words, the 2 different prosthesis types will form a prosthesis cluster within the hand–tool axis. We calculated a prosthesis-similarity index for each participant (see 'Methods' and S3 Fig for intercategory pairwise distances). We found a significant group difference of the prosthesis-similarity index ($t(54) = −2.55$, $p = 0.01$; Fig

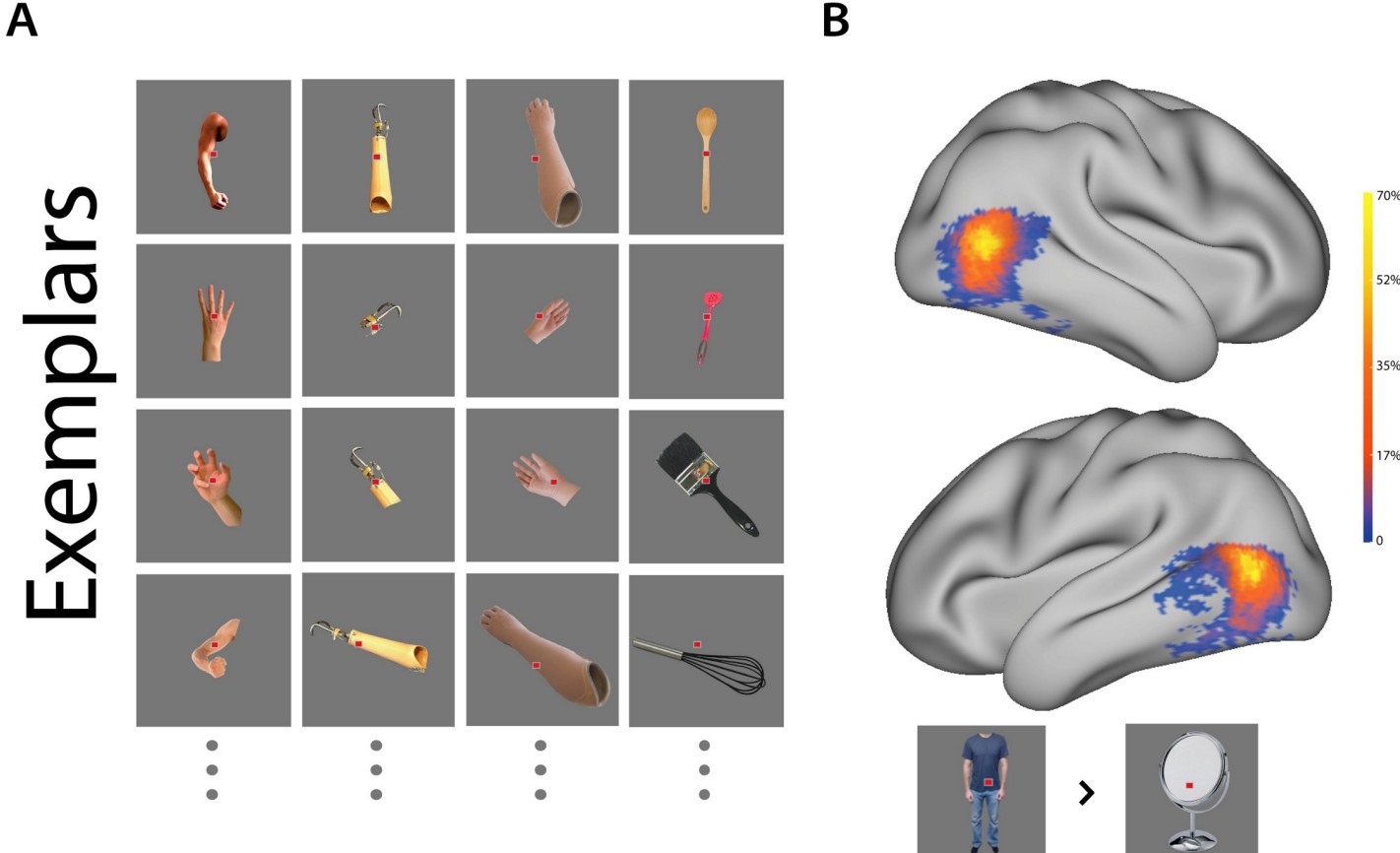

**Fig 1. Example stimuli and ROI.** (A) Example stimuli from the main 4 experimental conditions (columns, left to right): hands (upper limbs), active prostheses, cosmetic prostheses, hand-held tools. One-handers also observed images from multiple viewpoints of their own prosthesis. One image was shown per trial in an event-related design. (B) Probability maps of the body selective ROI. For each participant and hemisphere, the top 250 most activated voxels within the OTC were chosen based on a headless bodies > objects contrast, providing an independent ROI. ROIs from all participants (*n* = 56) were superimposed, yielding ROI probability maps. Warmer colours represent voxels that were included in greater numbers of individual ROIs. See S1 Fig for the probability maps of each group separately. Data used to create this figure can be found at https://osf.io/4mw2t/. ROI, region of interest.

3E). This indicates that using a prosthesis alters one's visual representation of different prostheses types into a distinct category and does so in a way that is more complex than prostheses simply becoming more hand- or tool-like. Although previous studies suggest tool representation is left-lateralised [42,49], the reported effect was robust across both hemispheres (right OTC: t(54) = −2.25 *p* = 0.03, left OTC: t(54) = −2.66, *p* = 0.01). Exploratory data-driven analysis ran on the one-handers pairwise distances comprising the prosthesis-similarity index further revealed that the departure of the active prostheses in particular away from its native 'tool' category is linked with its increased association with cosmetic prostheses (See S4 Table).

Supporting the interpretation that different neural representational structure in prosthesis users is associated with prosthesis usage, we found a significant positive correlation between the prosthesis-similarity index and the prosthesis usage score described above (Pearson's r(30) = 0.43, *p* = 0.01; Fig 3F). In other words, the more users use a prosthesis, the more they represented the different prostheses types as a separate category. Conversely, individuals who do not use a prosthesis frequently have a more similar representational structure to that of controls.

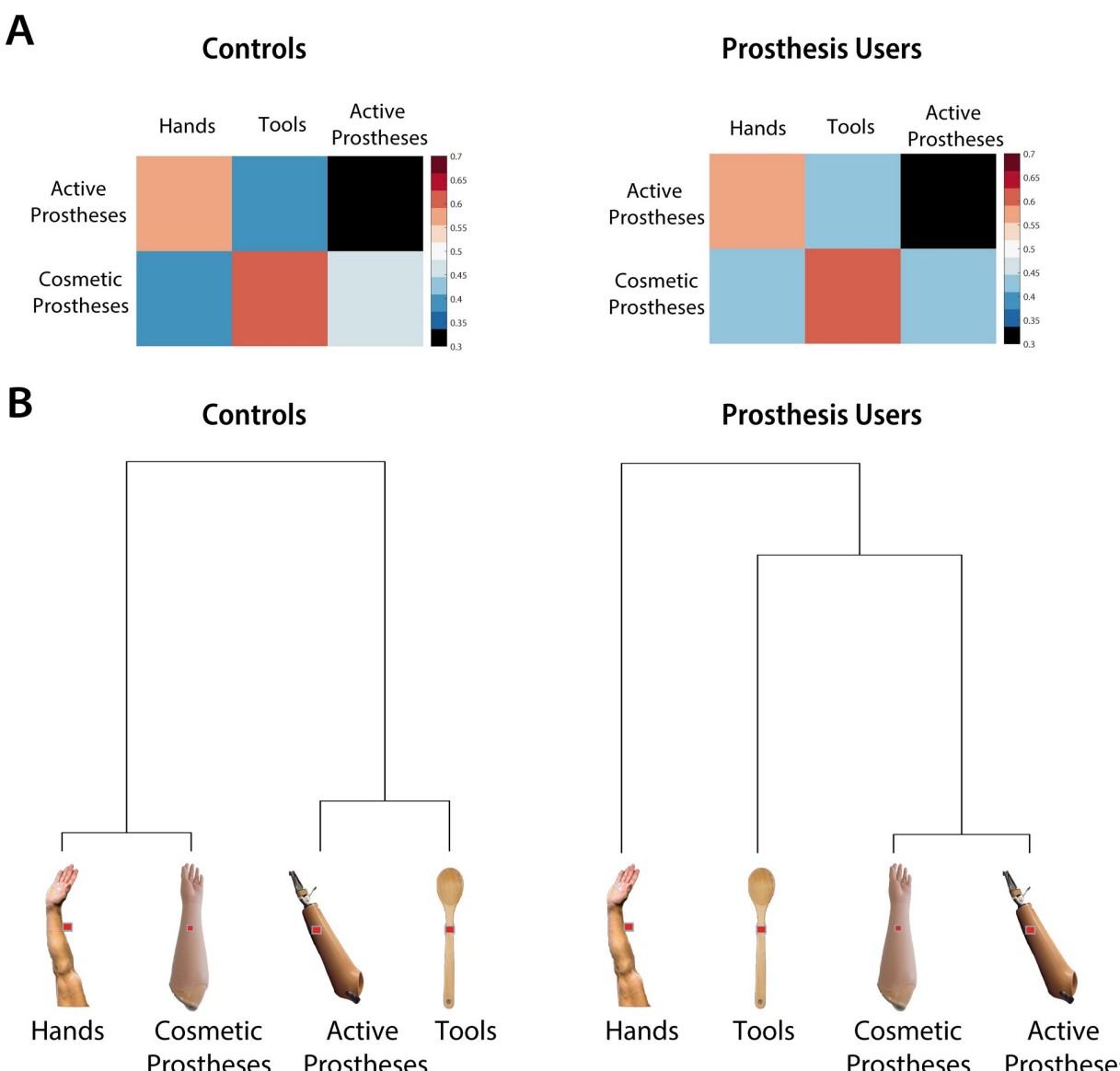

**Fig 2. Representational structure in body-selective visual cortex.** Multidimensional distances between activity patterns of each of the main condition (hands, tools, cosmetic prostheses, active prostheses) were calculated using representational similarity analysis. (A) Representational dissimilarity matrices for each group showing pairwise distances between the 2 prostheses conditions (active and cosmetic), hands, and tools. (B) To visualise the underlying representational structure a linkage tree (dendrogram) was calculated in each group of participants, combining information from all pairwise distances (two-handed controls, left; one-handed prosthesis users, right). Pairs of stimuli that are closer together in the multidimensional space are clustered together under the same branch. Longer connections across the vertical axis indicate greater relative distances. In controls, cosmetic prostheses are clustered with hands and active prostheses with tools, reflecting their visual similarities. In prosthesis users, however, the 2 prostheses types (cosmetic and active) are clustered together, with tools and hands represented dissimilarly from both prostheses. Data used to create this figure can be found at https://osf.io/4mw2t/.

The 2 indices for hand- and prosthesis-similarity are not statistically independent, and, as such, we were not able to compare them directly. However, we could use them as a model for predicting daily-life prosthesis usage. Comparing the correlation between prosthesis usage and the indices for the 2 hypotheses (embodiment and categorical) revealed a significant difference ($z = -2.52$, $p = 0.01$). Therefore, at least when considering unfamiliar prostheses, the prosthesis-similarity index is a better predictor of prosthesis usage than the hand-similarity index.

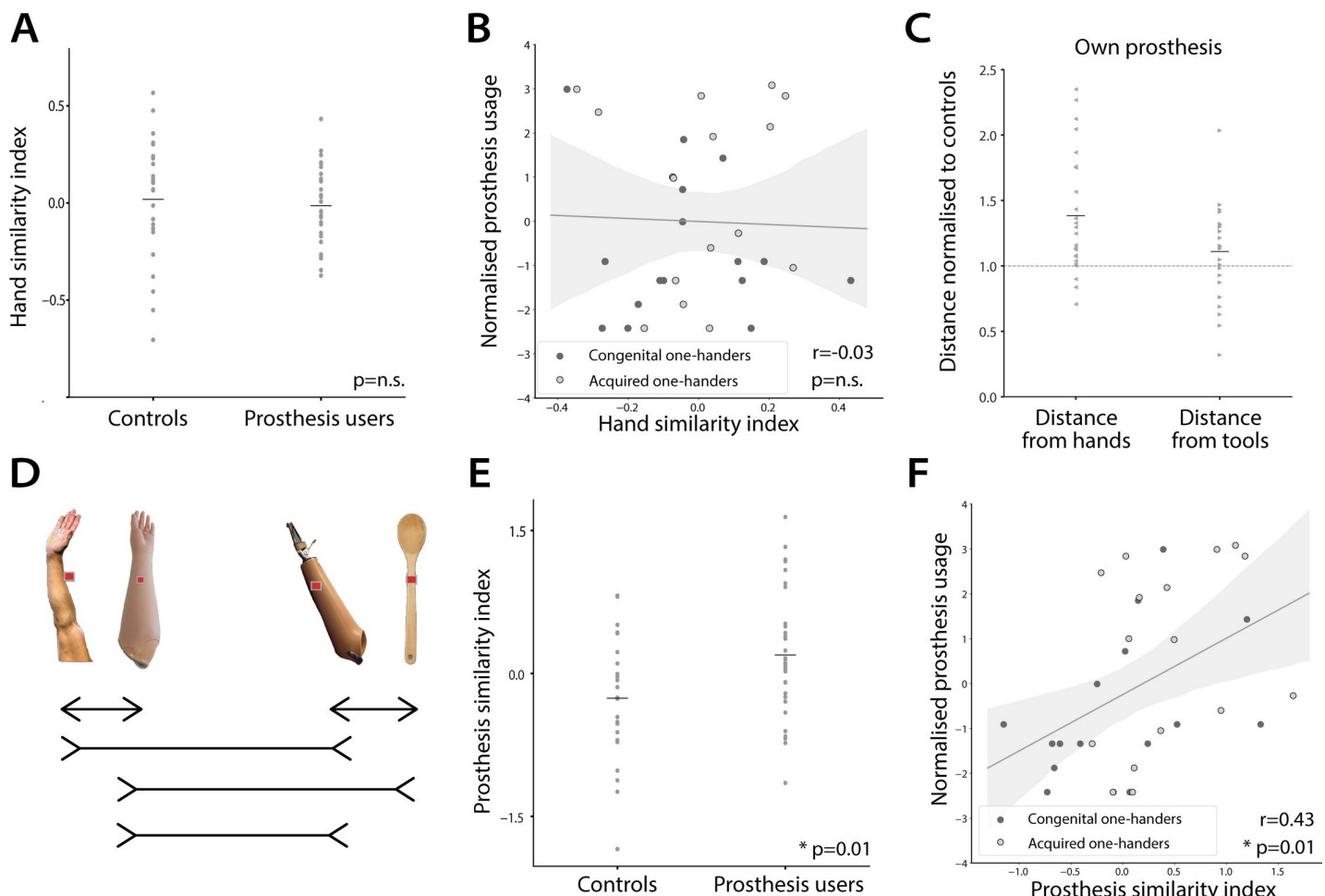

**Fig 3. Assessing the embodiment and categorisation hypotheses.** (A–B) A hand similarity index was calculated for each participant to quantify the degree to which both prostheses conditions (cosmetic and active) are more similar to hands than tools. A higher index in value indicates greater embodiment (hand similarity of prostheses). (A) A group comparison of the hand similarity index between controls and prosthesis users showed no significant differences (t(54) = 0.47, $p$ = 0.64). Horizontal lines indicate group means and dots indicate individual participants. (B) Correlation between the hand similarity index and prosthesis usage was not significant across users (Pearson's r(30) = −0.03, $p$ = 0.86). Dark/light circles indicate congenital/acquired one-handers, respectively, and grey shading indicates a bootstrapped 95% confidence interval. (C) Hand (left) and tool (right) distances from users' 'own' prosthesis. Individual distances were normalised by the controls' group mean distance, depending on the visual features of the 'own' prosthesis (hand likeness). A value of 1 indicates similar hand/tool distance to controls'. Users showed significantly greater distances between their own prosthesis and hands (t(25) = 4.33, $p$ < 0.001) contrary to the embodiment hypothesis. Together, these findings demonstrate that hand similarity under the embodiment hypothesis does not adequately explain differences in prosthesis representation in users' OTC. (D–F) A prosthesis similarity index was calculated for each participant to quantify the degree to which the prostheses representation moved away from their natural categories (hands for cosmetic prostheses and tools for active prostheses) and towards one another. (D) A visual illustration of the prosthesis similarity index formula. Arrows pointing outward indicate distances that should grow in users compared to controls (e.g., hands and cosmetic prostheses) and are therefore positively weighted. Arrows pointing inward indicate distances that should shrink in users compared to controls (i.e., cosmetic and active prostheses) and are therefore negatively weighted. (E) Group comparison of the prosthesis similarity index between controls, and prosthesis users showed greater prosthesis similarity in users (t(54) = −2.55, $p$ = 0.01). (F) The prosthesis similarity index significantly correlated with prosthesis usage; higher prosthesis index associated with greater prosthesis usage (based on wear frequency and skill; Pearson's r(30) = 0.43, $p$ = 0.01). Together, these findings demonstrate the categorisation hypothesis explains differences in user's prosthesis representation in the OTC, both with respect to controls and interindividual prosthesis usage. Data used to create this figure can be found at https://osf.io/4mw2t/. n.s., no significance; OTC, occipitotemporal cortex.

## Prosthesis categorisation does not depend on users' developmental period of hand loss or prosthesis type

When considering hand-selective neural resources, individuals who were born without a hand might develop different representational structures than those with an acquired amputation

[50]. Considering this, we tested whether the hand-similarity index differed between the 2 prosthesis users subgroups (congenital versus acquired) and found no significant differences (t(30) = −0.615, $p$ = 0.54). Moreover, the reported prosthesis-similarity effects prevailed even when accounting for any potential differences between the 2 subgroups, as demonstrated by an analysis of covariance (ANCOVA) with subgroup (congenital versus acquired) as a fixed-effect and usage as a covariate. The ANCOVA revealed no significant subgroup effect (F(1,29) = 2.02, $p$ = 0.17), demonstrating that prosthesis-similarity does not significantly differ because of cause/developmental period of hand loss, and a nearly significant usage effect (F(1,29) = 4.11, $p$ = 0.052), indicating that the relationship between prosthesis categorisation and usage is relatively independent from potential subgroup effects.

Beyond cause of limb loss, the users tested in this study also diverged with respect to prosthesis type, shape, and history of usage, involving primary users of cosmetic (40%), active (41%, comprising of mechanical hook [body-powered; 25%] and myoelectric [motor-powered, 16%]), as well as nonprosthesis users (16%) and a hybrid user (3%; see Table 1). A key question is what aspects of the prosthesis itself might affect neural prosthesis adoption in OTC. Because our key focus is on prosthesis usage, we looked at whether individuals who primarily use a prosthesis that has a degree of active grip control (e.g., a mechanical hook) show different effects for those who primarily use a cosmetic prosthesis that affords more limited motor control (no grip). Comparing the prosthesis-similarity index of users of the 2 prosthesis types revealed no significant effect (t(20) = 0.055, $p$ = 0.96). Using an ANCOVA with primary prosthesis type in daily life (cosmetic/active prosthesis) as a fixed effect and usage as a covariate revealed no prosthesis type effect (F(1,19) = 0.432, $p$ = 0.52), indicating that the categorisation effect might not depend on the type of prosthesis individuals primarily use. The usage effect remained significant (F(1,19) = 6.01, $p$ = 0.02), indicating that the correlation between usage and categorisation is independent of prosthesis type. Repeating the same analysis with the hand-similarity index revealed no significant effects, indicating that even when accounting for prosthesis type, no relationship is found with visual embodiment. Though null results should be interpreted with caution, our analysis indicates that the categorisation effect observed in prosthesis users is not driven by the prosthesis design or control mechanism but by the functionality the user extracts from it (as reflected in our daily usage scores).

In the active prosthesis condition, a minority of active prosthesis users ($n$ = 5) were shown images of a myoelectric prosthesis (that is not their own; Table 1 marked in asterisks). Because these are arguably visually distinct from the mechanical hooks seen by the control participants, we repeated the analysis of prosthesis-similarity index by replacing the subset of relevant pairwise distances relating to the active prosthesis with the mean distances of the prosthesis users' group (see 'Methods'). In this analysis, the observed effect remained but was reduced to a trend (t(54) = −1.87, $p$ = 0.067). Importantly, the correlation between categorisation and usage remained significant (r(30) = 0.48, $p$ = 0.006) even when excluding the myoelectric users from the analysis altogether (r(25) = 0.46, $p$ = 0.015). This further analysis confirms that our findings were not driven by the subset of myoelectric active prostheses.

## Own prosthesis representation

The results discussed so far were derived from visual presentation of prosthesis exemplars that each user was not personally familiar with, allowing us to easily compare the results between the users and controls. However, under the embodiment framework, it could be argued that embodiment can only be achieved for the user's own prosthesis. To account for this possibility, in addition to the general prosthesis types shown to all participants, most prosthesis users ($n$ = 26; see 'Methods' and Table 1) were also shown images of their own prosthesis (for the

many prosthesis users using more than one prosthesis, this refers to the prosthesis each user wore on the day of testing). Because controls do not have a prosthesis of their own, in this analysis, we compared the user's own prosthesis to the same prosthesis type shown to controls. Therefore, cosmetic 'own' prostheses ($n = 15$) were matched with controls' general cosmetic condition, and active 'own' prostheses ($n = 11$) were matched with the controls' general active condition. To allow us to group values across the cosmetic and active prostheses users, the distances between the 'own' prosthesis from hands and tools were normalised by the mean distances measured from the control group (using the above-mentioned conditions). Because we hypothesised that altered prosthesis representation is driven by usage, the controls' averaged distance is used here as a 'baseline' measure of how the representation is structured before prosthesis use. This normalised score was entered into a one-sample $t$ test for statistical comparison.

Based on the embodiment hypothesis, users should show greater similarity between users' own prosthesis and hands. Instead, we found that users showed significantly greater dissimilarity (greater distances), relative to controls, indicated by a normalised distance that was greater than 1 (t(25) = 2.85, $p = 0.009$). This analysis therefore shows that users' own prostheses are less similar to hands, providing direct evidence against the embodiment hypothesis. The normalised 'own' prosthesis distance from tools was also found to be significantly greater than 1 (t(25) = 2.91, $p = 0.008$), further supporting the categorisation interpretation. We also repeated the analysis as described above, but this time we standardised distances for 7 users with an 'own' active prosthesis with hand-like visual features (see Table 1) with controls' cosmetic prosthesis. This complementary analysis produced similar results for hands (t(25) = 4.33, $p < 0.001$) but not for tools (t(25) = 1.62, p = 0.118; Fig 3C). This means that even when taking both visual-feature and operational considerations, prosthetic limbs are not represented as hands in their owner's visual cortex.

## Prosthesis representation beyond EBA

To demonstrate that our results are not specific to our ROI definition in OTC of EBA, we have repeated the same analysis and found similar results in 'hand' and 'tool' ROIs within OTC, generated from the meta-analysis tool Neurosynth [51] (see S1 Text and S4 Fig).

We next explored prosthesis representation beyond OTC. The stimuli used in the current study were specifically designed to interrogate the information content underlying prosthesis representation in body-selective regions in the OTC. Nevertheless, as demonstrated in S5A Fig, the visual stimuli also significantly activated the intraparietal sulcus (IPS), providing us with an opportunity to explore visual prosthesis representation in a dissociated brain area (though notably, this activity was observed less consistently within individual subjects). Because 11 of the participants did not have enough significantly activated voxels within IPS to meet our ROI criteria, we constructed an anatomical ROI based on the Juliech Histological Atlas in FMRIB's Software Library (FSL), including human intraparietal (hIP)1-3 bilaterally [52,53] (see S5B Fig). With respect to the representational structure, we did not find significant differences between prosthesis users and controls when comparing both the hand- and prosthesis-similarity indices, even when wavering corrections for multiple comparisons, which are customary for exploratory analysis (hand-similarity: t(54) = 0.71, $p = 0.48$; prosthesis-similarity: t(54) = −0.45, $p = 0.66$). This could be due to insufficient power to explore this representational structure or possibly due to a different organising principle in this region. However, we did find that users showed significantly greater dissimilarity (greater distances), relative to controls, when comparing the representation of their own prosthesis to that of both hands and tools (hand: t(25) = 10.11, $p < 0.001$; tool: t(25) = 4.62, $p < 0.001$, corrected for 2 comparisons;

see S5C Fig). This analysis indicates that in parietal cortex, similar to the OTC, users' own prostheses are less similar to hands and tools, providing further support against the embodiment hypothesis.

## Discussion

Here, we used an fMRI brain decoding technique to probe the information content underlying visual prosthesis representation in the OTC. Previous research shows that prosthesis users are able to activate hand-selective areas in the OTC when viewing a prosthesis [29]. These areas, however, encompass a rich representational structure, spanning well beyond visual hand processing. It is therefore unknown whether, by utilising these hand-selective areas, the users' brain actually represents the prosthesis as a hand or whether it follows a different representational principle. We pitted the embodiment theory against another prominent theory—the categorisation theory—which is well established in visual neuroscience [43] (as detailed below) but to our knowledge has not been explored for wearable devices. Although not directly opposing, both theories generate different predictions on how a prosthesis should be represented in the brain of its user. Contrary to the embodiment theory, we found that prosthesis users represented their own prosthesis unlike a hand. For unfamiliar prostheses, users formed a prosthesis category, distinct from their natural (existing) categories (hands and tools), as demonstrated in controls. Importantly, this shift scales with usage, such that people who use their prosthesis more in daily life show more independence of the prosthesis category. When comparing the 2 models' success in explaining interindividual differences between prosthesis users, we find that the prosthesis category model was significantly more correlated with prosthesis usage. This indicates that, for visual representation of unfamiliar prostheses, categorisation provides a better conceptual framework than embodiment. Together with preliminary results showing that prosthesis users exhibited a less hand-like representation of their own prosthesis in parietal cortex, our results collectively show that neural visual embodiment does not predict successful adoption of wearable technology by the human brain. Despite benefiting from hand-selective cortical resources [29], the prosthesis is not 'embodied' into people's visual hand representation. We also did not find any evidence for greater representation of prostheses as tools. However, because the experimental design and analysis we implemented were a priori focused on visual hand representation in OTC, it is possible that other brain areas might find a different representational 'solution' to support the daily use of an artificial limb.

As stated above, an intuitive and increasingly popular view that has been inspiring biomedical design and innovation is that embodiment will optimise the usage of technology, such as upper-limb prostheses [10,11,13–16,54,55]. How can this view be resolved with our findings? One potential solution originates from the fact that embodiment is a multifaceted phenomenon [17], with distinct levels (i.e., phenomenological—does the prosthesis feel like my hand?; cognitive—do I react to/with the prosthesis like I would to/with my own hand?; neural—is the prosthesis represented via the same mechanisms as a hand? [7]). In another recent study in which we probed the phenomenological and cognitive levels of prosthesis embodiment, we found both to correlate with prosthesis usage [28]. Here, we focused our research on the neural level, because it is entirely unknown whether objective representation of the prosthesis as a body part associates with prosthesis acceptance and usage, let alone benefits it. It is possible, and even likely, that embodiment manifests differently in each of these distinct levels, which may also vary in their importance or even relevance for successful technological implementation [7]. To disentangle the complex concept of 'embodiment', future studies should aim to acquire measurements of the different levels of embodiment together with a detailed account of prosthesis skill and use.

A second important consideration is that embodiment is a multisensory process, involving multiple brain areas beyond visual cortex [41,56,57]. Here, we focused on visuomotor processing, and our experimental approach does have some limitations that should be considered. For example, our use of static images was designed to drive activity in the OTC but not in other sensorimotor-related regions in the parietal and frontal regions [58], thereby limiting our conclusions to high-level visual cortex. The use of generic hand images that are not the participants' own hands can also be limiting when approaching questions of embodiment. Here, it is important to mention that despite using a profoundly nonecological task in the scanner, the resulting brain representations, as captured with our prosthesis-similarity index, correlated significantly with the extent of everyday prosthesis usage. Therefore, despite the inherent limitations of our fMRI task, our task outcomes are ecologically relevant. Still, it is possible that other brain areas involved more directly in motor planning would produce other representational structures with respect to hand representation. Future research aimed at this question should take into consideration that at present, commercially available prosthesis motor control is fundamentally different from that of motor control of the hand, producing further potential challenges for sensorimotor embodiment.

Thirdly, and most speculatively, it is possible that the prosthesis may still be represented in OTC as a body part but one that is not a hand. After all, prosthesis users have strong semantic and sensorimotor knowledge of a hand (all users in the study had one intact hand; acquired amputees had prior experience of their missing hand, including lingering phantom sensations; see also the work by Striem-Amit and colleagues [59] for related findings showing normal visual hand representation in individuals born without both hands). Their experience with operating a prosthesis is fundamentally different from that of a hand. If body representation is not strictly tuned to the specific fine-grained features of a given body part (e.g., the digits of the hand) but is instead also represented at a higher level (e.g., effectors [60] or based on other functionality features [61]), then the dissociation of prostheses from hand representation observed in our study should not be taken as evidence for lack of embodiment per se but rather lack of embodiment as a hand. In this context, the previously reported recruitment of hand-specific visual cortical regions could reflect an underlying embodied representation of a prostheses as a (nonhand) body part. Therefore, we propose that future studies of artificial limb embodiment should not be limited to identifying and/or quantifying hand representation (as the current common practice, e.g., using the rubber hand illusion).

Instead of visual hand embodiment (or tool-like representation), we found that a significant amount of individual differences in prosthesis usage can be predicted by the extent of prosthesis categorisation within the visual body-selective area, providing a significantly better model than hand embodiment. This result is also consistent with the known organising principle of the OTC, in which categorical representation reflects our knowledge of grouping and patterns, which are not necessarily present in the bottom-up sensory inputs [43,62–64]. Moreover, categorical representation in OTC was shown to reflect individual differences in conceptual knowledge [65]. Accordingly, people who acquire a specific visual expertise, such as car experts, show increased activity in object-selective areas in OTC (for a review, see the work by Harel [66]). Research on object experts, therefore, provides compelling evidence for the role of visual learning and experience in shaping and fine-tuning categorical representation. Although various studies have demonstrated a relation between expertise and activation, few studies performed multivariate analyses, and those that did reported mixed results [44,67–70]. For example, Martens and colleagues, 2018, found that expertise did not alter the representational configuration of the category of expertise, whereas McGugin and colleagues, 2015, who studied car representation of experts in the Fusiform Face Area, report that car representation became more similar to that of faces with expertise. In this context, our present results provide

a novel perspective on visual expertise. This is because our results show divergence of prosthesis representation from the 'natural' categories normally existing in this brain area, consistent with the formation of a new categorical representation. In other words, rather than refining a category, prosthesis usage results in the formation of a new category. Gomez and colleagues, 2019, recently reported that childhood experience with playing a Pokémon videogame relates to increased classification of Pokémon, compared with other related categories, in the ventral occipital cortex [44]. Extending this finding, we report that categorical prosthesis representation correlates with (ongoing) visuomotor experience in adulthood. As these effects were found in both congenital and acquired one-handers, this prosthesis categorisation effect does not seem to relate to a specific developmental period. Because prosthesis usage relies on visuomotor expertise, it is difficult for us to disentangle the relative contribution of perceptual expertise to the observed prosthesis categorisation. Further research examining the representation of prosthesis in exclusively perceptual experts (e.g., prosthetists) will provide an interesting test case for our expertise hypothesis.

A further distinguishing feature of prosthesis users compared to other experts is that they not only have increased experience with prosthetic limbs but also, arguably, a reduction in exposure to hands, at least from a first-person perspective. Reorganisation is the process by which a specific brain area qualitatively changes its input–output dynamics to support a novel representation. This raises the question of whether or not the OTC becomes reorganised to support a new visual function (prosthesis control, known to strongly rely on visual feedback [18,71]). In this context, in recent years, adaptive behaviour has been suggested [60,72–75] and challenged [59,76,77] as a causal driver of brain reorganisation. According to this framework, the function of the deprived cortex is not reassigned; instead, it is the input (prosthesis versus hand) that changes, while the local processing and resulting output persist (domain specificity [78,79]). For example, in a study conducted in individuals with a congenitally missing limb, multiple body parts used to compensate for the missing limb's function benefited from increased activity in the missing hand's sensorimotor cortical area [60], replicated in the recent work by Hahamy and Makin [80]. It was, therefore, suggested that opportunities for reorganisation may be restricted by a predetermined functional role of a given brain area, e.g., hand resources will only support processing related to hand-substitution behaviours (other body parts or a prosthesis). This framework has been successfully implemented to demonstrate that the categorical organisation of OTC is preserved following congenital blindness [81–83]. For example, the OTC body area was shown to be selectively activated by tactile [84] and auditory [85] inputs conveying hand/body information. Our findings advance beyond these studies by demonstrating that a parallel form of reorganisation can occur even when the relevant sensory pathway (vision) is largely unaffected, further highlighting the role of daily behaviour in shaping brain organisation across the life span.

Finally, our results suggest that the relationship between prosthesis representation and usage is independent of key design and usage features of the artificial device (such as visual mimicry of the human hand) and cause of limb loss (congenital or acquired). This should inspire future efforts in neurologically-informed substitution and augmentative artificial limb design to not be strictly confined to biomimetics, a design principle that is currently highlighted in the development of substitution technology [86] (e.g., the vine prosthesis [87]).

To conclude, we provide a novel neural correlate for the adoption of wearable technology that is distinct from visual embodiment of a prosthesis as a hand. Successful prosthesis usage, in terms of both wear time and habit in daily life, was predicted not by visual embodiment (hand-similarity) or tool-similarity but by a more distinct categorical representation of artificial limbs. Understanding whether the brain can treat a prosthesis as a hand and whether this hand-like representation provides a real advantage for prosthesis users will have important

implications on future design and assessment of wearable technology. Considering the limitations related to our focus on visual prosthesis representation in passive settings, we are currently unable to offer a sweeping answer to how the entire brain represents artificial limbs. However, our findings provide an important alternative to the highly prominent embodiment theory that needs to be considered. As such, much more research is necessary to provide a comprehensive understanding of the neural basis of successful prosthesis usage in the human brain.

## Methods

### Ethics statement

This study was approved by the Oxford University's Medical Sciences interdivisional research ethics committee (Ref: MSD-IDREC- C2-2014-003). Written informed consent and consent to publish was obtained in accordance with ethical standards set out by the Declaration of Helsinki.

### Participants

Thirty-two individuals missing an upper-limb (one-handed prosthesis users, mean age [SD] = 42.3 [11.8], 12 females, 8 missing their right hand) were recruited to take part in the study (Table 1). Sixteen prosthesis users lost their hand following an amputation, and sixteen had a unilateral congenital upper-limb below-elbow deficiency (due to complete arrest of hand development). One additional one-hander was recruited to the study but did not participate in the scanning session because of claustrophobia. In addition, 24 age- and gender-matched two-handed controls (age = 41.7 [13.1]; 12 females; 8 left-handed) took part in the study. Fourteen of the control participants were family members, friends, or held professional relationships with prosthesis users, resulting in passive visual experience of prosthesis usage. All participants took part in a larger study, involving multiple tasks (https://osf.io/kd2yh/). Univariate data from the fMRI task reported here was previously published [29].

### Prosthesis usage measurements

Prosthesis usage was assessed by combining two, highly correlated measurements of usage: prosthesis wear frequency and a prosthesis activity log (PAL) [29,88]. Participants rated their prosthesis wear frequency on a scale: 0 = never, 1 = rarely, 2 = occasionally, 3 = daily ($<$4 hours), 4 = daily (4–8 hours), 5 = daily ($>$8 hours). Some participants use more than one type of prosthesis; in that case, the measurement from the most frequently used prosthesis was used. The PAL is a revised version of the motor activity log [89] as described in the work by Makin and colleagues [74]. In brief, participants were requested to rate how frequently (0 = never, 1 = sometimes, 2 = always) they incorporate their prosthesis in an inventory of 27 daily activities, with varying degrees of motor control. PAL is calculated as the sum of the levels of frequencies in all activities divided by the maximum possible sum ($27 \times 2$) creating a scale of 0 to 1. This questionnaire, indexing bimanual usage, was previously validated using limb acceleration data [74] and behavioural lab testing [60], collected in ecological settings. Because neither measure is able to fully capture prosthesis use, both the prosthesis wear frequency and PAL were Z-transformed and summed to create a prosthesis usage score.

### Stimuli

Participants viewed still object images of the following categories: (i) hands (upper limbs with and without the arm, in multiple orientations and configurations, and of different sizes,

genders, and skin-colours; hereafter hands); (ii) man-made hand-held tools; (iii) cosmetic prostheses (aesthetically hand-like but nonoperable), (iv) active prostheses (affording a grip; either mechanical hooks or myoelectric prostheses); and (v) (when available) participants' own prosthesis (more details below). For hands and prosthesis images, the effector was matched to the prosthesis users' missing-hand side or the nondominant hand in controls (e.g., participants missing their left hand were presented with 'left-handed' hands/prostheses). Headless bodies and typically nonmanipulable man-made object images were also included for localising independent ROIs (see "OTC ROI" section). Additional conditions that were also included in the fMRI paradigm and initial analysis but not reported here were dismorphed images (originally intended to account for low-level visual activity but discarded after univariate analysis revealed increased activity in OTC) and lower limbs (intended as a control body part but not included in final analysis).

Images used for the main stimulus categories can be found at https://osf.io/kd2yh/. All images had their background removed, normalised for size, placed on an equiluminant grey background, and overlaid with a red fixation square. Nonprosthesis conditions were taken from an online database and involved multiple exemplars, whereas each of the 3 prosthesis conditions involved multiple shots of a single prosthesis of different orientations. We chose to use multiple configurations of hands and prostheses to probe the experience of congenital one-handers and control participants (who mostly see prostheses/hands-of-the-missing-side from a third person perspective, respectively). We note that previous studies probing visual hand representation in similar populations [88], including specifically in OTC [59], used a similar approach. We further note that the few studies finding differences between egocentric/allocentric [90] or self/others [91] visual hand representation in OTC identified lateralised effects to the right OTC, whereas our effects are comparable across hemispheres. Prosthesis images of other users' prostheses (in the cosmetic and active conditions) or of the participant's prosthesis (in the 'own' condition) were taken by the experimenters prior to the functional MRI session. The subset of the active prosthesis users (marked with an asterisk in Table 1) were shown another myoelectric prosthesis in the active prosthesis condition.

In the 'own' prosthesis condition, all prosthesis users who had brought their prosthesis to the study were presented with images of their own prostheses, either cosmetic or active (*n* = 26; see Table 1). All other participants (i.e., the remaining 6 prosthesis users who did not bring a prosthesis and all control participants) were shown pictures of their own shoe instead. Shoes were selected as a familiar external object that was intended to exert similar cognitive effects (e.g., in terms of arousal) as the prosthesis and therefore minimise differences in the scan time course across groups. Because we had no a priori interest in studying shoe representation, the shoe condition was not included in further analysis.

Post hoc shape similarity analysis [92] confirmed that the prosthesis images spanned a diverse range, resulting in similar shape dissimilarity for the 2 prosthesis types with respect to hand and tool exemplars (S2 Fig). It is highly likely that other measurements of visual similarity (e.g., based on contrast/colour comparison or perceptual judgements) would reveal more distinct intercategorical (dis)similarities. However, any such visual (dis)similarities should impact intercategorical representational similarity in the control group. As such, in the present study, all key analyses are interpreted with respect to the controls, providing us with a representational 'baseline'.

## Experimental design

Each condition consisted of 8 different images. In each trial, a single image from one of the conditions was shown for 1.5 s, followed by 2.5 s of fixation. Participants were engaged in a

one-back task and were asked to report whenever an image was repeated twice in succession. This occurred once for each condition within each functional run, resulting in 9 trials per condition per run (8 distinct exemplars and 1 repetition). This design was repeated in 4 separate functional runs, resulting in 36 events per condition. First-order counterbalancing of the image sequences was performed using Optseq (http://surfer.nmr.mgh.harvard.edu/optseq), which returns the most optimal image presentation schedule. Run order was varied across participants. The specifics of this design were validated against an event-related design with a jittered interstimulus interval and a block design during piloting (*n* = 4). Stimuli were presented on a screen located at the rear end of the scanner and were viewed through a mirror mounted on the head coil. Stimulus presentation was controlled by a MacBook Pro running the Psychophysics Toolbox in MATLAB (The MathWorks, Natick, MA).

## MRI data acquisition

The MRI measurements were obtained using a 3-Tesla Verio scanner (Siemens, Erlangen, Germany) with a 32-channel head coil. Anatomical data were acquired using a T1-weighted magnetization prepared rapid acquisition gradient echo sequence with the parameters: TR = 2040 ms, TE = 4.7 ms, flip angle = 8˚, and voxel size = 1 mm isotropic resolution. Functional data based on the blood oxygenation level-dependent signal were acquired using a multiband gradient echo-planar T2$^*$-weighted pulse sequence [93] with the parameters: TR = 1300 ms, TE = 40 ms, flip angle = 66˚, multiband-facto = 6, voxel size = 2 mm isotropic, and imaging matrix = 106 × 106. Seventy-two slices with slice thickness of 2 mm and no gap were oriented in the oblique axial plane, covering the whole cortex, with partial coverage of the cerebellum. The first dummy volume of each scan was saved and later used as a reference for coregistration. Additional dummy volumes were acquired and discarded before the start of each scan to reach equilibrium. Each functional run consisted of 256 volumes.

## Preprocessing and first-level analysis

fMRI data processing was carried out using FEAT (FMRI Expert Analysis Tool) version 6.00, part of FSL (www.fmrib.ox.ac.uk/fsl). Registration of the functional data to the high resolution structural image was carried out using the boundary based registration algorithm [94]. Registration of the high resolution structural to standard space images was carried out using FLIRT [95,96] and was then further refined using FNIRT nonlinear registration [97,98]. The following prestatistics processing was applied; motion correction using MCFLIRT [96]; nonbrain removal using BET [99]; B0-unwarping using a separately acquired field-map; spatial smoothing using a Gaussian kernel of FWHM 4 mm; grand-mean intensity normalisation of the entire 4D data set by a single multiplicative factor; highpass temporal filtering (Gaussian-weighted least-squares straight line fitting, with sigma = 50 s). Time-series statistical analysis was carried out using FILM with local autocorrelation correction [100]. The time series model included trial onsets and button presses convolved with a double gamma HRF function; 6 motion parameters were added as confound regressors. Indicator functions were added to model out single volumes identified to have excessive motion (>1 mm). A separate regressor was used for each high motion volume, no more than 9 volumes were found for an individual run (3.5% of the entire run).

## OTC ROI

Because the focus of the study was on visual representation of prostheses in the OTC, the representational similarity analysis was restricted to individualised ROIs. ROIs of the extrastriate body area (EBA) [47] were identified bilaterally in each participant, by selecting the top 250

activated voxels, in each hemisphere, in the headless bodies > nonmanipulable objects. Voxel selection was restricted to the lateral occipital cortex, inferior and middle temporal gyri, occipital fusiform gyrus, and temporal occipital fusiform cortex (all bilateral), as defined by the Harvard-Oxford atlas [101]. Voxels from both hemispheres were treated as a single ROI (see S3 Table for the all analyses repeated for each hemisphere separately). Mean activity within the ROI for each participant was calculated by averaging the parameter estimate (beta) for each condition across all 500 voxels.

### Representational similarity analysis

To assess the hand–prosthesis–tool representation structure within the ROI, pairwise distances between conditions were calculated using a multivariate approach, generally known as representational similarity analysis [102]. Prior to performing the multivariate analysis, we first examined differences in univariate activity in the ROI, which could drive differences in the multivariate analysis. When comparing activity levels (averaged across all voxels) between controls and prosthesis users within this region, no significant differences were found for each of the image conditions of hands, tools, and prostheses ($p > 0.1$ for all; see S2 Table), indicating that the 2 groups did not activate this region significantly differently. We then continued with the multivariate analysis. For each participant, parameter estimates of the different conditions and GLM residuals of all voxels within the ROI were extracted from each run's first-level analysis. To increase the reliability of the distance estimates, parameter estimates underwent multidimensional normalisation based on the voxels' covariance matrix calculated from the GLM residuals. This was done to ensure that parameter estimates from noisier voxels will be downweighted [48]. Cross-validated (leave-run-out) Mahalanobis distances (also known as LDC—linear discriminant contrast) [48,103] were then calculated between each pair of conditions. Analysis was run on an adapted version of the RSA Toolbox in MATLAB [103], customised for FSL [104]. Visualisation of the distances in dendrogram was performed using the plotting functions available in the RSA Toolbox.

### Hand-similarity and prosthesis-similarity indices

Two indices were calculated to test each of the aforementioned predictions—one based on embodiment and the other on the categorical structure of OTC. Each index's formula was designed so that positive values support the prediction. Therefore, similar to GLM contrasts, each of the relevant pairwise distances were weighed with a positive multiplier if, under the specific prediction, that distance should grow (decreased similarity) in prosthesis users, and a negative multiplier if it should shrink (greater similarity). For instance, the embodiment hypothesis predicts that for both prostheses (active and cosmetic) the distance from tools would grow (will be less similar), whereas the distance from hands will shrink (will be more similar). Therefore, the formula used to calculate the Hand-Similarity index was (Tools↔CosmP + Tools↔ActP) − (Hands↔CosmP + Hands↔ActP), in which '↔' indicates the distance between a pair of conditions. Using the same logic, the prosthesis-similarity index was calculated as a measurement of how much the 2 prosthesis conditions are represented as a separate cluster away from their native condition (cosmetic prostheses resembling hands and active prostheses resembling tools). The index was calculated using the following formula: 3 (Hands↔CosmP + Tools↔ActP) − 2(Hands↔ActP + Tools↔CosmP + ActP↔CosmP); see Fig 3D for a visualisation of the formula. To control for individual differences in absolute distance values, both indices were standardised by the individuals' distances between hands and tools (i.e., the residuals after accounting for the variance in the Hands↔Tools distance; see S3 Table for all analyses performed with the raw index values).

As mentioned earlier (see 'Stimuli'), 5 active prosthesis users viewed images of myoelectric prostheses as the active prosthesis condition while all other participants viewed mechanical hooks under that condition. Because this creates a possible bias in our analysis, we have attempted to remedy this in two ways. The first is to replace all distances that involved the active prosthesis condition with the mean distances of the rest of the users' group. In other words, before calculating the indices, replacing the distances (Tools↔ActP, Hands↔ActP, ActP↔CosmP) for these 5 individuals with the mean distances of the remaining 27 users. Another approach was to remove these individuals from the analysis altogether.

## Analysis by prosthesis type

To test the influence of prosthesis type on users' prosthesis-similarity index, we repeated the same analysis as described above and compared the hand- and prosthesis-similarity indices between individuals using a cosmetic prosthesis ($n = 13$) and active prosthesis users ($n = 9$). The following participants have been excluded from this analysis: (1) 5 individuals not using a prosthesis (usage time of 0 = never); (2) the 5 participants that viewed myoelectric prostheses as the active prosthesis condition mentioned above.

## Own prosthesis representation

In this analysis we aimed to assess each user's own prosthesis representation, defined by its distance from hands and tools. Our approach was designed to overcome 2 challenges: First, controls do not have an 'own' prosthesis; and second, the visual and operational features of different prostheses may vary significantly and need to be accounted for before similarity measures can be averaged across all prosthesis users. Therefore, for each participant, we normalised (divided) the 'own' prosthesis distance by the mean distance of a similarly looking/operating prosthesis as found in controls. This produced a measure that reflects the magnitude of the representational shift of individual's 'own' prosthesis from the average distance of the control participants. A value of one would therefore indicate no difference between controls and the user's representation of their own prosthesis. For the 7 prostheses users wearing an active prosthesis that had hand-like visual features (see Table 1 for a full breakdown of prostheses types), we repeated the analysis twice over, standardising their distances with control's active and cosmetic prosthesis distances. This allowed us to account for both visual and operational features.

## IPS ROI

The IPS ROI was generated using the Juliech Histological Atlas, including all voxels that have more than 30% probability of being within the grey matter of the IPS areas: hIP1, hIP2, and hIP3, [52,53] in both hemispheres.

## Statistical analysis

All statistical testing was performed using SPSS (version 24), with the exception of the Bayesian analysis which was run on JASP [105]. Comparisons between prosthesis users and two-handed controls were performed using a two-tailed Student $t$ test. To test the relationship between the indices and individuals' prosthesis use, a usage score, described above, was correlated with the indices using a two-tailed Pearson correlation. An ANCOVA with prosthesis usage as a covariate was used to test the contribution of several factors such as cause of limb loss and type of prosthesis used. Own prosthesis analyses were performed using a one-sample $t$ test, comparing the mean to one as the control means were used to calculate the individual indices. To

interpret key null results, we ran a one-tailed $t$ test. The Cauchy prior width was set at 0.707 (default). We interpreted the test based on the well accepted criterion of Bayes factor smaller than 1/3 [106,107] as supporting the null hypothesis. Corrections for multiple comparisons were included for exploratory analysis when resulting in significant differences, as indicated in the results section. To minimise flexible analysis, which could lead to p-hacking [108], only a limited set of factors, prespecified in the initial stages of the analysis or based on the reviewer's comments, were tested in this manuscript. Further exploratory analysis on other recorded clinical factors can be conducted using the full demographic details: https://osf.io/kd2yh/.

## Supporting information

**S1 Text. Supplementary results: 'Hand' and 'Tool' ROI analysis.** Additional analyses conducted on 'hand' and 'tool' ROIs generated from the meta-analysis tool Neurosynth. ROI, region of interest
(DOCX)

**S1 Table. Control participants demographics.** This table was taken from the supplementary material of van den Heiligenberg and colleagues, 2018, using the same controls cohort.
(XLSX)

**S2 Table. Group comparison of average activity levels.** Group comparison of average activity levels between controls and prosthesis users within the visual body-selective ROI. Results are shown for both the bilateral ROI (500 voxels) and for each hemisphere separately (250 voxels each). ROI, region of interest
(XLSX)

**S3 Table. Confirmatory additional analyses.** A summary table for analyses of hand-similarity index and prosthesis-cluster index: group comparisons (controls versus prosthesis users) and correlation with prosthesis usage. Including the results reported in the paper (bilateral ROI), results within the same ROI with the raw indices without controlling for the hand-tool distance (bilateral ROI raw), and for the results of the indices for each hemisphere separately. ROI, region of interest
(XLSX)

**S4 Table. PCA of pairwise distances.** To explore which of the pairwise distances contributed to the underlying observed effect of prosthesis categorisation we ran a data-driven analysis (PCA) on the 5 distances of the one-handed group. Values in the table are the weights given to each distance within a component. The first component shows a 'main effect' of interindividual differences across participants, in which some individuals have overall larger distances than others across all condition pairs. In our calculated indices, we control for this effect by normalising the individual's selectivity indices by their Hands ↔ Tools distance (see 'Methods'). The second component explains almost half of the remaining variance (after accounting for the interindividual differences in component 1). In the second component, individuals showing greater distances between the active prostheses and the tool condition also show greater similarity between the active prosthesis and the cosmetic prosthesis conditions. In other words, when the active prosthesis condition moves away from the tool category, it also tends to get closer to the cosmetic prostheses (as can be seen by the high weights and opposite signs of these 2 distances in the second component). This data-driven analysis provides further support for the hypothesised categorical shift of prosthesis representation. PCA, principle component analysis
(XLSX)

**S1 Fig. Group probability maps for visual body-selective ROIs in control participants and prosthesis users.** All individual visual ROIs were superimposed per group, yielding corresponding probability maps. Warmer colours represent voxels that were included in greater numbers of individual ROIs. Data used to create this figure can be found at https://osf.io/4mw2t/. ROI, region of interest
(TIF)

**S2 Fig. Intercategorical shape dissimilarities.** (A) Two exemplars from the 'hand' and 'active prosthesis' categories. (B) All exemplars shown to each individual participant were submitted to a visual shape similarity analysis (Belongie and colleagues, 2002), in which intercategorical pairwise shape similarity was assessed. (C) A histogram showing intercategorical similarity from one participant's shown cosmetic (blue) and active (red) prosthesis exemplars, with respect to hand exemplars (all exemplars are available on https://osf.io/kd2yh/). As demonstrated in this example, these dissimilarity ranges were largely overlapping. (D) This intercategory dissimilarity analysis was repeated for each of the participants (based on the specific prostheses exemplars shown to them), and mean histogram values were averaged. As indicated in the resulting matrix, cosmetic and active prostheses did not show strong differences in similarities, on average. This is likely due to the wide range of exemplars/shapes used in the study data set. Data used to create this figure can be found at https://osf.io/4mw2t/.
(TIF)

**S3 Fig. Pairwise distances in EBA.** (A) Pairwise distances between patterns of activations of hands, cosmetic prostheses, active prostheses, and tools. In the labels, '↔' indicates the distance between a pair of conditions. Within the plot, x indicates the group's mean. (B) same as panel A, only with each distance standardised by the individuals' distances between hands and tools. (C) A table illustrating the direction of the effect predicted by each index. Data used to create this figure can be found at https://osf.io/4mw2t/. EBA, extrastriate body-selective area
(TIF)

**S4 Fig. Neurosynth 'hand' and 'tool' ROIs.** Using the association maps for the words: 'hand' (blue) and 'tools' (green), ROIs were defined by using all significant voxels within the OTC. These ROIs are projected on inflated brain for visualisation. Surface and volume masks can be found at https://osf.io/4mw2t/. OTC, occipitotemporal cortex; ROI, region of interest
(TIF)

**S5 Fig. IPS analysis.** (A) Univariate activations in prosthesis users. Results of the group level univariate contrast of (Active Prosthesis + Cosmetic Prosthesis) > Objects show that at the group level the IPS is also activated. (B) The IPS region of interest was taken from the Juelich Histological Atlas (30% probability of hIP1, hIP2, and hIP3). (C) Hand (left) and tool (right) distances from users' 'own' prosthesis in IPS. Individual distances were normalised by the controls' group mean distance, depending on the visual features of the 'own' prosthesis (hand-likeness). A value of 1 indicates similar hand/tool distance to controls. Users showed significantly greater distances between their own prosthesis and hands ($t(25) = 10.11$, $p < 0.001$) contrary to the embodiment hypothesis. A significant increase in the distance of the 'own' prosthesis from tools was also observed ($t(25) = 4.62$, $p < 0.001$). Data used to create this figure can be found at https://osf.io/4mw2t/. hIP, human intraparietal; IPS, intraparietal sulcus.
(TIF)

## Acknowledgments

We thank the authors of our previous manuscript [29] and, in particular, Fiona van den Heiligenberg and Jody Culham for contributions in experimental design, data collection, and helpful advice on data analysis. We thank Opcare for help in participant recruitment; Chris Baker and Stephania Bracci for their comments on the manuscript; and our participants and their families for their ongoing support of our research.

## Author Contributions

**Conceptualization:** Tamar R. Makin.

**Data curation:** Tamar R. Makin.

**Formal analysis:** Roni O. Maimon-Mor.

**Methodology:** Roni O. Maimon-Mor.

**Supervision:** Tamar R. Makin.

**Validation:** Roni O. Maimon-Mor.

**Visualization:** Roni O. Maimon-Mor.

**Writing – original draft:** Roni O. Maimon-Mor, Tamar R. Makin.

**Writing – review & editing:** Roni O. Maimon-Mor, Tamar R. Makin.

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
