## [Editor Report · Decision Letter 0]

15 Jul 2019

Dear Dr Makin, 

Thank you for submitting your manuscript entitled "Is an artificial limb embodied as a hand? Brain decoding in prosthetic limb users" for consideration as a Research Article by PLOS Biology.

Your manuscript has now been evaluated by the PLOS Biology editorial staff, as well as by an Academic Editor with relevant expertise, and I am writing to let you know that we would like to send your submission out for external peer review.

Please note, however, that we would like to pursue your manuscript as a Short Report, not as a Full Research Article. Please select this option when resubmitting.

In addition, before we can send your manuscript to reviewers, we need you to complete your submission by providing the metadata that is required for full assessment. To this end, please login to Editorial Manager where you will find the paper in the 'Submissions Needing Revisions' folder on your homepage. Please click 'Revise Submission' from the Action Links and complete all additional questions in the submission questionnaire.

**Important**: Please also see below for further information regarding completing the MDAR reporting checklist. The checklist can be accessed here: https://plos.io/MDARChecklist

Please re-submit your manuscript and the checklist, within two working days, i.e. by Jul 17 2019 11:59PM.

Kind regards,

Gabriel Gasque, Ph.D.,

Senior Editor

PLOS Biology

INFORMATION REGARDING THE REPORTING CHECKLIST:

PLOS Biology is pleased to support the "minimum reporting standards in the life sciences" initiative (https://osf.io/preprints/metaarxiv/9sm4x/). This effort brings together a number of leading journals and reproducibility experts to develop minimum expectations for reporting information about Materials (including data and code), Design, Analysis and Reporting (MDAR) in published papers. We believe broad alignment on these standards will be to the benefit of authors, reviewers, journals and the wider research community and will help drive better practise in publishing reproducible research. 

We are therefore participating in a community pilot involving a small number of life science journals to test the MDAR checklist. The checklist is intended to help authors, reviewers and editors adopt and implement the minimum reporting framework. 

IMPORTANT: We have chosen your manuscript to participate in this trial. The relevant documents can be located here:

MDAR reporting checklist (to be filled in by you): https://plos.io/MDARChecklist

**We strongly encourage you to complete the MDAR reporting checklist and return it to us with your full submission, as described above. We would also be very grateful if you could complete this author survey:

https://forms.gle/seEgCrDtM6GLKFGQA

Additional background information:

Interpreting the MDAR Framework: https://plos.io/MDARFramework

Please note that your completed checklist and survey will be shared with the minimum reporting standards working group. However, the working group will not be provided with access to the manuscript or any other confidential information including author identities, manuscript titles or abstracts. Feedback from this process will be used to consider next steps, which might include revisions to the content of the checklist. Data and materials from this initial trial will be publicly shared in September 2019. Data will only be provided in aggregate form and will not be parsed by individual article or by journal, so as to respect the confidentiality of responses. 

Please treat the checklist and elaboration as confidential as public release is planned for September 2019.

We would be grateful for any feedback you may have.

---

## [Decision Letter · Decision Letter 1]

28 Aug 2019

Dear Tamar,

Thank you very much for submitting your manuscript "Is an artificial limb embodied as a hand? Brain decoding in prosthetic limb users" for consideration as a Short Report at PLOS Biology. Your manuscript has been evaluated by the PLOS Biology editors, by an Academic Editor with relevant expertise, and by four independent reviewers. Please accept my apologies for the delay in sending the decision below to you.

In light of the reviews (below), we will not be able to accept the current version of the manuscript, but we would welcome resubmission of a much-revised version that takes into account the reviewers' comments. We cannot make any decision about publication until we have seen the revised manuscript and your response to the reviewers' comments. Your revised manuscript is also likely to be sent for further evaluation by the reviewers.

Your revisions should address the specific points made by each reviewer. Having discussed these comments with the Academic Editor, we think you should pay very special attention to the common concern about using visual areas and passive assessments. For example, is there evidence of similar findings in motor areas. If you chose to send in a revision, this point should be clearly and fully addressed.

Please submit a file detailing your responses to the editorial requests and a point-by-point response to all of the reviewers' comments that indicates the changes you have made to the manuscript. In addition to a clean copy of the manuscript, please upload a 'track-changes' version of your manuscript that specifies the edits made. This should be uploaded as a "Related" file type. You should also cite any additional relevant literature that has been published since the original submission and mention any additional citations in your response. 

Before you revise your manuscript, please review the following PLOS policy and formatting requirements checklist PDF: http://journals.plos.org/plosbiology/s/file?id=9411/plos-biology-formatting-checklist.pdf. It is helpful if you format your revision according to our requirements - should your paper subsequently be accepted, this will save time at the acceptance stage.

Please note that as a condition of publication PLOS' data policy (http://journals.plos.org/plosbiology/s/data-availability) requires that you make available all data used to draw the conclusions arrived at in your manuscript. If you have not already done so, you must include any data used in your manuscript either in appropriate repositories, within the body of the manuscript, or as supporting information (N.B. this includes any numerical values that were used to generate graphs, histograms etc.). For an example see here: http://www.plosbiology.org/article/info%3Adoi%2F10.1371%2Fjournal.pbio.1001908#s5.

For manuscripts submitted on or after 1st July 2019, we require the original, uncropped and minimally adjusted images supporting all blot and gel results reported in an article's figures or Supporting Information files. We will require these files before a manuscript can be accepted so please prepare them now, if you have not already uploaded them. Please carefully read our guidelines for how to prepare and upload this data: https://journals.plos.org/plosbiology/s/figures#loc-blot-and-gel-reporting-requirements.

Upon resubmission, the editors will assess your revision and if the editors and Academic Editor feel that the revised manuscript remains appropriate for the journal, we will send the manuscript for re-review. We aim to consult the same Academic Editor and reviewers for revised manuscripts but may consult others if needed.

We expect to receive your revised manuscript within two months. Please email us (plosbiology@plos.org) to discuss this if you have any questions or concerns, or would like to request an extension. At this stage, your manuscript remains formally under active consideration at our journal; please notify us by email if you do not wish to submit a revision and instead wish to pursue publication elsewhere, so that we may end consideration of the manuscript at PLOS Biology.

When you are ready to submit a revised version of your manuscript, please go to https://www.editorialmanager.com/pbiology/ and log in as an Author. Click the link labelled 'Submissions Needing Revision' where you will find your submission record. 

Sincerely,

Gabriel Gasque, Ph.D., 

Senior Editor

PLOS Biology

Reviewer remarks:

Reviewer #1: The authors present a study on the fMRI activation in the occipitotemporal cortex (OTC) related to different prostheses types. They found that prosthesis-use let to OTC fMRI activations more similar to those of tools than to hands, and that greater daily-life prosthesis usage correlated with greater prosthesis (tool) categorisation, documenting use-dependent plasticity within the OTC. Overall, the manuscript is well written, the methods clear and the results interesting, but there a few issues that need to be addressed. 

- It is stated that “Two hypotheses were tested: the embodiment hypothesis, assessing whether prosthetic limbs are in fact represented as hands (and not tools); and, the categorization hypothesis, assessing whether a new ‘prosthesis’ category has formed.” It is unclear what predictions these hypotheses make about the presented outcome measures (representational similarity as measured as OTC fMRI activation). Please clarify in the introduction. 

- "In other words, prosthesis users do not, on average, ‘embody’ prostheses as hands more than controls." The lack of a statistically significant difference in the hand-similarity index in prosthesis users relative to controls is not a good argument in favor of the equality of this measure, i.e., an argument against the embodiment hypothesis. Please provide equivalence test to support this statement, or at least a power analysis.

- What was the rationale for choosing to examine BOLD activity in the extra striate region of the OTC? What about motor cortex representations? Were any other regions examined? What were the findings?

- The lack of a linear relationship between prosthesis use and the hand similarity index is contrasted against the strong linear relationship between prosthesis use and the prosthesis similarity index. The latter relationship doesn't need to be driven by the separation of prosthesis from hand representations, because the former relationship would have reflected this too. This implies that the strong linear relationship between ‘prosthesis use’ and ‘prosthesis similarity index’ is driven by the distance between prosthesis representations. Please disentangle the distance between prosthesis representations from the distance between other prosthesis and hand representations.

- "Therefore, our results show that neural embodiment does not predict successful adoption of wearable technology by the human brain." The authors found that prosthesis use correlates negatively with the prosthesis similarity index (“the degree to which the prostheses representation moved away from their natural categories (hands for cosmetic prostheses and tools for active prostheses) and towards one-another”). Doesn't this imply a negative correlation between neural embodiment (representational separation from hands) and adoption of the technology (prosthesis use)?

- "Successful prosthesis usage was predicted not by embodiment (hand-similarity), 

but by a more distinct categorical representation of artificial limbs." What do you mean by successful prosthesis use? How did you measure prosthesis success rate? Unclear, please reword or explain.

Reviewer #2: In this study Mor and Makin use multivariate pattern analysis of BOLD signals in the occipitotemporal cortex (OTC) to compare the neural representations of visually presented hands and prostheses in prosthesis-users and controls. The main result is that protheses are more clearly represented as its own category in prosthesis-user compared to controls. In other words, the OTC can better distinguish between hands and prostheses in the group that uses protheses. This is an interesting observation that is well worth reporting. The paper is well written, the statistical analyses has been conducted appropriately, and the main result is very clear. The study advances our understanding of OTC, use-dependent plasticity and the neural representation of artificial limbs in higher order visual cortex. However, I think the authors overstate importance of the present finding with respect to theories of embodiment and the development of advanced prostheses. Other weaknesses are that not all conclusions in the discussion are supported by the data and the manuscript includes some unnecessary speculations. Further, the finding that a new 'prothesis' category is formed in the OTC is not entirely novel as we know from earlier studies that expertise can lead to changes in the neural representations of categories in OTC. In sum, this is an interesting high-quality study but, in my opinion, the conceptual advance with respect to our understanding of artificial limb embodiment is somewhat limited. This article might therefore be better suited in a more specialized journal. 

Major points. 

1. In the discussion I think the authors oversell their results as evidence against the embodiment hypothesis. The authors do discuss some limitations of their methods and results but I still think they go too far in their conclusions. The present study does not investigate fronto-parietal-cerebellar regions during actual prothesis use so the embodiment theory has not been directly tested in my opinion. To me it seems reasonable that the OTC in expert prothesis-users should be better at differentiating between prosthetic limbs and real hands compared to controls. But does this really matter for how visual information about the prosthesis is processed by parietal, frontal and cerebellar regions during prosthesis use? Take hand-held tools for example. Clearly the OTC can differentiate body parts from tools but tools can be embodied in the sense that some of the same sensorimotor mechanisms used to control real limbs are re-used for effective control of hand-held tools. So, my point is simply that I think the authors should have a more balanced discussion with respect to the significance of the present results for theories of prothesis embodiment. 

2. The authors do discuss some limitations of the study with respect to the embodiment question, most notably the use of 2-D pictures of hands rather than real protheses and the focus on a single visual area rather than a network of sensorimotor regions. This is all very good, but to me a further limitation is that the authors used hands and prostheses depicted from different visual orientations, and most problematically, mixing up first person perspective and third person perspective. Does this not cause a problem for the interpretation of the results? Hands (or protheses) of strangers viewed from a third person perspective should not be embodied but simply treated as external visual object. How could this mixing of perspectives have influenced the current results? 

3. I think the authors needs to work more on the last paragraph of the discussion. I see two problems with this “conclusion paragraph”. The first is that the authors suddenly introduce a new third interpretation of their data (“appendage represented as a body part”). This seems out-of-place here in the conclusion paragraph. If this is an important alternative interpretation – which I think it is – then it should be discussed more extensively earlier in the discussion paragraph. Also, the authors should more explicitly state how this “appendage interpretation” relates to the two hypotheses outlined in the introduction and how it fits with the results more precisely. But if this idea is merely a speculation it might be better to drop it all together. The second problem I had with the “conclusion paragraph” is that the authors argue against using rubber hand illusion as a tool for prosthesis embodiment research. I do not follow. The present study did not investigate the rubber hand illusion or the sense of ownership of prosthetic limbs, so to me this part reads like an unnecessary speculative overstatement. In sum, I recommend a shorter conclusion-paragraph without speculations or introduction of new concepts and interpretations.

Minor points

4. In my opinion, the statement “In other words, prosthesis users do not, on average, ‘embody’ prostheses as hands more than controls.” is an overstatement. This conclusion is based on a non-significant result from a frequentist statistical test. In other words, all you can say is that you did not find significant evidence against the null hypothesis. But this does mean that you can conclude that the null hypothesis is true. Please tone down this statement or present results from a Bayesian statistical analysis that support the null hypothesis. (This comment refers to a sentence in the results paragraph with the heading “Prosthesis-like (categorical), and not hand-like (embodied) representation of prostheses in

prosthesis users”)

5. In the methods section (under “Stimuli”) please add a motivation for mixing the presentations of the pictures of the hands and protheses from the first- and third-person perspectives (also see point 2 above). Please also clarify which perspective that was used for the own hand stimuli. Finally, add an explanation for why the shoe condition was not included in the analysis. 

6. It would have been interesting to have ratings regarding how much the amputees embodied their prothesis in every-day life. I assume that according to the embodiment hypothesis this should have some relationship to the neural representation of protheses. If I am not misinformed, most prothesis users describe their prothesis more like a tool than an actual hand. 

7. In the second discussion paragraph you mention the sensorimotor related regions in the frontal and parietal regions, but give no references. Please add a couple of references here. 

8. I miss page numbers in the manuscript.

Reviewer #3: The authors present an interesting study aiming at characterising how the visual cortex, and more specifically the extra striate body area, reacts to images of prostheses, tools and hand in prosthesis users and controls.The topic and objective of the paper, understanding how the brain embodies technologies -here a prosthetic hand-, is not only important conceptually but also has major implication for rehabilitation since technological development is not sufficient for successful device adoption and usage. Indeed, to better understand how amputees adopt their prosthesis, one needs to measure objectively embodiment. The authors decided to have an implicit measure of embodiment based on recording the activity of region in the visual system known to typically show specific responses to tools and body-parts like hands. This study is an important extension of a previously published paper showing that individuals who used their prosthesis more in daily lives also showed greater activity in the OTC’s hand-selective visual areas when viewing images of prostheses (van den Heiligenberg, Brain, 2018). The main claims of the paper is straightforward: in contrast to what is predicted to an embodied view of brain (re)organization in amputees, the authors show that the upper limb prothesis is represented in the occipital cortex differently that the hands. The paper is clearly written and the study was elegantly designed and expertly executed. Authors should be praised to rely on fMRI data of 32 individuals missing a hand, which is a sample size larger than typically reported in the literature. Here follows some suggestion that may help to improve the study.

The authors set out to explore two alternative hypotheses (embodiment vs categorisation hypotheses) by mainly contrasting brain activity evoked by prostheses versus hand. However an important third hypothesis is not fully tested in my view: that prostheses are represented not like hand but like tools. This third hypothesis presume that prostheses representations will anchor on an existing category (tools) which is different from what the authors suggest, namely the creation of a new categorical distinction due to new conceptual (visual? -see below) knowledge. 

As discussed by the authors, the main finding (as summarised above) could well be explained by visual expertise. Alternatively, since it was shown that body-selective regions also link to the parietal cortex to represent body and tool actions, one may think that the organisation of this brain region also depends on the use of the prosthesis. At the moment the paper does not disentangle these two alternatives which would tell us important information about the mechanisms underlying the brain reorganisation in amputees and people born without hand. Would it be possible to find a strategy to disantagle those two alternatives by analysing further the brain representation of your own prosthesis versus another prosthesis with similar design-function? In this vein, it might also be interesting to scan the brain of people involved in rehabilitative training of amputees since one can assume similar visual expertise without the motor component of use. I think also that this topic (visual vs user-dependent expertise) should be put upfront in the paper in the introduction.

Authors focused their study on body selective regions. Why not conducting similar analyses on more specific ROIs selective to hands or tools? I understand that body selective regions could be defined independently (univariate analyses) from the conditions of interest undergoing multivariate analyses, which may not be the case for hands/tools. I am however wondering if the author could use a data split-half strategy (or any type of crossvaligdation technique) or relying on hand and tool related ROIs from the literature to achieve this goal?

In a previous study, the authors found that individuals who used their prosthesis more in daily lives also showed greater activity in the OTC’s hand-selective visual areas when viewing images of prostheses (van den Heiligenberg, Brain, 2018). Here, they find that when comparing activity levels between controls and prosthesis- users within this region, no significant differences were found for each of the image conditions of hands, tools and prostheses . How do these two observations made on the same dataset fit?

A group comparison of the hand-similarity index between controls and prosthesis users showed no significant differences. In addition, users showed significantly greater distances between their own prosthesis and hands. Both results are thought to contrast with the embodiment hypothesis but the link between those two results is unclear to me.

The prosthesis categorisation effect was found in both congenital and acquired one-handers suggesting no "sensitive period" for this effect to emerge. However the analyses in link to the comparison of these two groups could be further developed in the manuscript.

The analysis were implemented a priori focusing on prosthesis and hand representation in the occipital cortex. Do the authors have speculation about what is happening in other brain regions like the sensorimotor cortex which is thought to also activate in link to action related objects- it is possible that a different representational structure support the use of an artificial limb in those regions? 

The study included additional conditions like dismorphed images, lower limb and the own shoes (for people that were shown picture of their own prosthesis). What was the rational for not analysing these conditions which could provide important additional insights?

Distance measurement between brain patterns were computed using Cross validated Mahalanobis distances. Is it what is referred to as linear discriminant contrast (Walter 2016)?

The main hypotheses were tested by a series of group comparisons using t-tests. How were they corrected for multiple comparisons?

Reviewer #4: The manuscript is well written and describes an interesting set of data that challenges a widely-accepted viewpoint in the field, which is that upper limb prostheses are embodied as natural hands. The manuscript offers a new view, which is that prostheses are neurally represented as distinct from both hands and other tools. The manuscript also reports on prosthesis usage data and correlates prosthesis usage to neural representations. The manuscript discusses an important and timely topic, and has implications both for rehabilitation and human-machine interfacing.

Major Points:

1. My primary concern with the study is the underlying assumption that neural activity in visual areas of the brain is an appropriate indicator of embodiment. The study attempts to extend findings about neural responses in the occipitotemporal cortex (OTC) as evidence that prostheses are not embodied as a hand. However, to my knowledge, embodiment is generally discussed in the literature in the context of the body schema, which is a preconscious sensorimotor representation of the body. I would expect then, that neural responses in the sensory and motor cortices would be more appropriate to measure embodiment-related neural representations of prostheses, tools, and hands. Why did the authors choose to focus on OTC rather than any sensorimotor brain regions? I am not convinced that neural responses in visual brain areas are sufficient indicators of embodiment, and thus am not convinced that the conclusions are supported by the data. Please comment.

2. In addition, given that sensorimotor representations of the body involve active control of a limb or tool and dynamic sensory feedback from the limb or tool, why was the experimental task designed to study passive visual experience rather than control or somatosensory feedback of the presented objects? I am not convinced that passively viewing an image of a prosthesis is a sufficient indication of the neural representation of the prosthesis during actual prosthesis usage. Please comment.

3. I believe that the perspective of the body-related images may have impacted the definition of the ROI and the measured neural responses to the hands and tools. One does not typically view one’s own body from a third-person perspective, but the study included both first- and third-person perspective images. What was the rationale for including third-person images when the goal was to assess embodiment, which relates to the sense of self? For the images shown in the third-person, I am concerned that the neural responses recorded might be related to viewing other people’s bodies rather than one’s own body. Were there any differences in neural representations between images in the first- and third-person perspective? This seems to me to be a critical point. Furthermore, in Figure 1, I cannot distinguish which of the example images are in the first-person perspective because they are all floating in a solid background disconnected from bodies. Are the authors certain that the images were interpreted by the participants as being in the first person? Please comment.

4. The survey results regarding prosthesis usage are interesting and valuable. However, I believe that grouping these two metrics, frequency of wear and PAL, may not be appropriate. Users of cosmetic prostheses, for example, may wear their prostheses very frequently, but may not use them to perform any tasks. Although the authors note that the PAL has been validated, they do not present any indication that this combined prosthesis usage score has been validated. Could the authors please provide information about the validation of this combined usage metric? I would also be interested to see the analysis comparing hand-similarity index to prosthesis usage broken down for the two components of the usage metric: i.e. one plot of hand-similarity index to wear frequency and one plot of hand-similarity index to PAL score. Did the authors attempt to decouple the two usage metrics?

5. I found the discussion about visual expertise to be very interesting in relation to the conclusions of this study. It seems to me that the emergence of prosthesis-specific visual expertise in prosthesis users is well-supported by the results, and perhaps a better interpretation of the data than a lack of embodiment. The authors discuss prior literature on car experts, who show increased activity in object-selective areas of OTC compared to non-experts. In this context, the prosthesis users could be considered prosthesis experts, since they have more experience with prostheses than control participants (perhaps except in rare cases). If car-specific regions of the OTC emerge through visual learning, it is reasonable that prosthesis-specific regions could also emerge through visual learning. If there were a ‘prosthesis-specific area’ of the OTC, then it could follow that prosthesis experts would have increased activation of this area than non-experts. This prosthesis-specific region of the OTC could be considered to be the non-hand, non-tool region the authors observed in their fMRI data. It would also make sense that prosthesis usage would correlate with the emergence of the prosthesis-specific region in the OTC, since visual learning would be promoted by increased visual exposure to prostheses (as the prosthesis is used more over time). However, I do not think that this visual learning interpretation implies that prostheses are not embodied, because visual learning could occur independently from sensorimotor learning, and to my understanding, sensorimotor learning is intricately related to embodiment.

Minor Points:

1. The manuscript needs to be revised throughout for typos.

2. I am concerned about the way in which the ROI was defined based on the presentation of headless-bodies versus everyday objects. Why were the bodies headless? Could the unnaturalness or dehumanizing nature of the images of headless-bodies impact arousal or emotion-related centers of the brain?

3. For viewing images of prostheses, was the handedness of the prosthesis images matched to the handedness of the prosthesis user’s limb loss? For example, for a person with an amputation of the right arm, were the observed images of right-handed prostheses? Could the authors please comment on how handedness may have impacted the results?

4. In the introduction, the authors suggest that prosthetic limbs are the “most established form of wearable technology to date.” While I agree that prosthetic limbs are an important category of wearable technology, many other wearable technologies, such as hearing aids, glasses, and watches, are much more widely adopted than prostheses.

5. In the Discussion, the authors note that the categorization hypothesis is “well validated in neuroscience” but do not present any citations to demonstrate prior instances of this categorization hypothesis in the literature. Please provide citations for this prior work.

---

## [Decision Letter · Decision Letter 2]

4 Feb 2020

Dear Tamar,

Thank you very much for submitting a revised version of your manuscript "Is an artificial limb embodied as a hand? Brain decoding in prosthetic limb users" for consideration as a Short Report at PLOS Biology. This revised version of your manuscript has been evaluated by the PLOS Biology editors, and by the original Academic Editor and reviewers. You will note that reviewer 4, Emily Graczyk, has signed her comments.

In light of the reviews (below), we are positive about your paper and pleased to offer you the opportunity to address the remaining points from the reviewers in a revised version that we anticipate should not take you very long. We will then assess your revised manuscript and your response to the reviewers' comments and we may consult the reviewers again.

We expect to receive your revised manuscript within 1 month. 

**IMPORTANT - SUBMITTING YOUR REVISION**

Your revisions should address the specific points made by each reviewer. You will see that while most of these lingering concerns/comments require only textual modifications (more discussion, some clarifications), some need more analyses/data, particularly reviewer 1’s point about Fig S3, reviewer 2’s point 3.1, and reviewer 4’s points 2 and 4. Having discussed these comments with the Academic Editor, we think you should provide what the reviewers have requested, with exception of reviewer 2's point 3.1, which will be your choice. 

Because this is a Short Report, please keep the final number of main figures below four. 

Please submit the following files along with your revised manuscript:

*Resubmission Checklist*

*Published Peer Review*

*PLOS Data Policy*

*Blot and Gel Data Policy*

Sincerely,

Gabriel Gasque, Ph.D., 

Senior Editor

PLOS Biology

REVIEWS:

Reviewer #1: The authors have provided extensive replies and significantly expanded on the original manuscript in the first round of corrections. They have convincingly made the claim that their "categorization hypothesis" (that prostheses become their own semantic category) provides a better explanation than the traditional embodiment hypothesis (that protheses are represented more like hands). At the same time, the selected paradigm does support complete falsification of the embodiment hypothesis. fMRI activation patterns in the OTC during passive viewing of prostheses may not be the optimal measure to make claims about the embodied representation that would invariably involve widespread brain areas including the motor and somatosensory cortices during actual use of a prosthesis. Therefore, conclusions about the explanatory power of the embodiment hypothesis should be toned down.

A few more comments:

Currently, cosmetic and active prostheses are lumped together in most of the outcome measures. It could be the case that the embodiment hypothesis holds true only for one of them. 

Supplementary Figure S3 is greatly appreciated to better understand the individual pairwise distances that make up the complex prosthesis similarity index which underlies the core argument of the manuscript - that the categorization hypothesis provides a better explanation of brain activity patterns than the embodiment hypothesis. Please provide statistical hypothesis tests for each of these displayed pairwise distances, and present this information in the main text. It is quite confusing to interpret the prosthesis similarity index without understanding the distances it is composed of. As the prosthesis similarity index is a key outcome measure, this is of paramount importance.

If the prosthesis similarity index should capture "degree to which the prostheses representation moved away from their natural categories (hands for cosmetic prostheses and tools for active prostheses) and towards one another", why does it include the pairwise distances hands <-> active prostheses and tools <-> cosmetic prostheses? It doesn't appear to be directly relevant under the above definition, and confuses the reader during their interpretation of the measure. 

Reviewer #2: Excellent revision. The manuscript has been much improved and it will now make a very valuable contribution to the literature. The results are fascinating. I just have a few minor comments.

1. Introduction, page 2. The term "visual embodiment" is new to me. Can we you define the term and/or give a reference the first time you use it on page 2. 

2. Introduction, page 2. The phrase "visual embodiment has been suggested to the gateway for the sense of bodily ownership" is jargon. Please provide a better explanation for why visual information is important for embodiment and body perception. From a multisensory perceptive vision is important for localizing and identifying the limbs in space because the signal is spatially more precise compared with other senses (under good viewing conditions). I recommend you cite studies from the multisensory literature. 

3. In their response letter, the author write that some authors have argued that the rubber hand is not embodied at a sensorimotor level during the rubber hand illusion, but rather at the visual level. I disagree with this view and argue that most researchers working on the rubber hand illusion probably describe this phenomenon as a multisensory body illusion rather than a visual illusion. The somatosensory (tactile-proprioceptive) aspects of the illusion are very prominent and the illusion works well without visual feedback of a limb (Ehrsson et al. 2005 J Neurosci; Guterstam et al. 2013 J Cog Neurosci). 

4. Page 3 and discussion. Most importantly, when the authors discuss OTC and EBA it is probably relevant to mention that this region respond to the rubber hand illusion, over and above tactile stimulation or the presentation of a visual hand image. The two best citations are probably Limanowski and Blankenburg (2014) Neuroimage (see also Limanowski and Blankenburg 2016 J Neurosci) that used an EBA localizer, and Gentile et al. (2013) J Neurosci that showed a correlation between LOC activity and the strength of the ownership illusion and who also found significant changes in functional connectivity between LOC and IPS (stronger PPI during hand ownership). This literature does not only suggest that OTC/EBA respond to tactile stimulation and has cross-modal properties, but actually, that this region play an important role in the key multisensory integration mechanisms related to body ownership/embodiment. I think this point is relevant for current study. 

5. Page 14, first sentence, second paragraph. What is the difference between "neural visual embodiment" and "visual embodiment"? You have not registered "visual embodiment" behaviorally in this study, so we have no data about a possible relationship between visual embodiment and prothesis-use from a behavioral perspective. Please clarify this section. 

6. Page 14, fourth paragraph, "other sensorimotor-related regions in the parietal and frontal regions (53)". I recommend that the authors to cite a couple of relevant original articles here (e.g. Gentile et al 2013), and a more up-to date-review, for example Ehrsson (2020). 

Ehrsson HH. Multisensory processes in body ownership. In: Sathian K, Ramachandran VS, eds. Multisensory Perception: From Laboratory to Clinic. Academic Press: Elsevier; 2020:179-200. 

http://www.ehrssonlab.se/pdfs/Ehrsson%20HH%202020%20Multisensory%20processes%20in%20body%20ownership.pdf

Reviewer #3: The authors have done a meticulous job addressing my comments. I think a few minor issues may require further considerations.

The number below relates to my original comments.

3.1.I believe it would be better to formalise the 3rd possibility (that prosthesis anchor more into the representation of tools in prosthesis user) from the start- eg in the intro. As mentioned by the author this hypothesis is 'implicitly" announced and tested and I see that the results is confusing as others reviewers mention that prosthesis maps more like tools while the author in their response suggest that this is not statistically the case. Also, I think it makes sense to consider this possibility seriously since one of the main hypothesis for the development of tool selective region relate to tool-use and intrinsic connectivity between occipital-temporal and parietal regions (e.g. work of Stefania Bracci). I however let the author decide if they think this proposed strategy is best suited to present their work.

3.2. I understand that the authors do not want to include these exploratory analyses in the manuscript. I would however be nice for the reader to get familiarised in the discussion with the idea that the visual expertise hypothesis, which is one potential reason behind the results of the current study, might be formally tested for instance by involving people highly involved in rehabilitative training of amputees.

3.4. I am not sure the point was fully taken. What I was intending to question was the conceptual relationship of the current paper in the context of the Brain paper. 

In the abstract of the Brain paper, one can read: "We show that the more one-handers use an artificial limb (prosthesis) in their everyday life, the stronger visual hand-selective areas in the lateral occipitotemporal cortex respond to prosthesis images." or, in the summary section "our findings show that neurocognitive resources devoted to representing our body can support representation of artificial body parts". This paper therefore suggest that images of prostheses anchor into visual hand selective region in prosthesis user. At first sight, this seems at odd with the conclusion of the current study showing that "prosthesis users represented their own prosthesis more dissimilarly to hands, compared to controls". I think that some text unifying the two studies may help more naive reader on the subject to create a coherent core of knowledge in this domain.

Reviewer #4, Emily Graczyk: I have reviewed a previous version of the manuscript. I believe the current version is much improved and I thank the authors for their efforts to thoroughly address the points raised by reviewers. I have a few remaining comments that I would like the authors to consider.

Major points:

1. I thank the authors for clarifying that the focus of the article is on visual embodiment rather than sensorimotor embodiment. I think this clarification alleviates most of my previous major concerns. The authors' explanation of why visual embodiment was more appropriate to test in this study rather than sensorimotor embodiment was also helpful.

As another argument for the relevance of visual embodiment in one-handers, the authors note the over-reliance on visual guidance during normal prosthesis use. Could the authors comment on how this over-reliance on visual feedback when using prostheses could have influenced the results? Is it possible that the over-reliance on visual information could be a factor contributing to the differences in cortical representations of prostheses for prosthesis users compared to controls? 

2. On page 8, the authors state "This indicates that using a prosthesis alters one's visual representation of different prostheses types into a distinct category, in between hands and tools…" It appears that the authors assume that the prosthesis category must be in between hands and tools along the hand-tool axis. In a multidimensional space, however, it's possible for the prosthesis category to be off of the hand-tool axis in a higher dimension. The calculations of the hand-similarity index and the prosthesis-similarity index do not appear to necessitate a single axis along which all three categories (hand, tool, prosthesis) must lie. I believe this implicit assumption of a single axis could influence the interpretation of the study by readers and should be removed. As a way to address this point, it would be interesting to see an analysis breaking down the hand similarity index and the prosthesis similarity index into their component distance contributors. For example, how much of the differences between controls and prosthesis users in the prosthesis similarity index are due to the hand-cosmetic distance increasing vs the tool-active prosthesis distance increasing vs the cosmetic-active distance decreasing? Based on the way the prosthesis similarity index is calculated, any one of these distance changes could result in a higher prosthesis similarity index without all three distances necessarily changing. A discussion about the contributions of the inter-condition distances to the similarity indices and the extent to which the prosthesis category is on or off the hand-tool axis would be helpful. 

3. The authors discuss the present findings in relation to two of their prior studies that appear to present contradictory or unexpected conclusions. The authors should provide a bit more discussion about how to rectify or interpret these different findings. First, the authors cite a prior study in the introduction showing that greater prosthesis use in daily life correlated to greater activity in OTC. However, the first paragraph of the results indicates that overall activity levels in the selected ROI were not different between prosthesis users and controls. This seems to contradict the previous results cited in the introduction and warrants further discussion. In addition, what are the implications of greater overall activation of OTC with increased prosthesis use on the inter-group comparisons in this study? Second, the authors discuss a prior study showing that cognitive and phenomenological embodiment correlates to prosthesis usage. While this does not contradict the present findings that neural embodiment does not correlate to prosthesis usage, it does suggest that cognitive embodiment is not a consequence of neural embodiment in the visual cortex. Instead, it suggests that cognitive embodiment must arise from neural representations in some other brain area. A bit more discussion about these points would be helpful in interpreting the present work in the context of prior literature.

4. I believe an analysis comparing the data from congenital and acquired one-handers is needed to support the authors' claim that the relationships do not depend on "visual mimicry of the human hand and cause of limb loss" (page 17). It appears that the data is presented separately for congenital and acquired one-handers in Figures 3 and 4, but there do not appear to be any statistical analyses presented that explicitly compare the two groups of one-handers. Were there any differences between congenital and acquired one-handers in hand similarity index or prosthesis similarity index? Were the relationships between prosthesis similarity index and prosthesis usage the same for congenital and acquired amputees? 

5. I think the manuscript was improved by the additional analyses focused on ROIs outside of EBA, and I thank the authors for their efforts on this. However, I think the interpretations stated in the section "Prosthesis representation beyond EBA" are stated too strongly. The only significant differences shown in IPS between prosthesis users and controls was for the "own prosthesis" condition. The hand and prosthesis similarity indices did not differ between controls and prosthesis users in the IPS. While I do not think this point weakens the overall study, I think it should be made clear that prosthesis categorization may be weaker in IPS. I think this is further support for future investigations into neural representations of embodiment in other brain areas, as the authors suggest in the discussion.

Minor points:

1. I thank the authors for clarifying why the task was designed to be passive. However, I have a few remaining questions about the images viewed by participants while in the scanner. The authors state that they used generic hand images for the task. Please clarify if the hands were matched in size, gender, and/or skin color to the participants' own hands. I think this detail needs to be explicitly stated in the methods or supplement, so that readers can understand the point the authors make in the discussion regarding the limitations of "generic hand images that are not the participants' own hands".

2. I thank the authors for the additional information about the combined prosthesis usage measure and the additional analyses demonstrating the relationships between the similarity indices and the PAL and wear time individually. In the previous revision, I asked the authors to provide evidence that the combined prosthesis usage measure had been validated. By validation of the metric, I mean is there data demonstrating internal consistency, minimal detectable change, and test-retest reliability of this metric? If this data is available, please indicate where it can be found. If not, please indicate this point in the methods.

3. In the abstract, the sentence beginning with "Moreover, prosthesis users represented their own prosthesis more dissimilarly to hands, compared to controls, challenging…" should be reorganized for clarity. It is difficult to distinguish what exactly was compared.

4. Page 12 - the acronym RSA is introduced and not defined.

5. Where was the prosthesis observation log (POL) score for control participants (shown in Table S1) used in this study? I do not think Table S1 should be removed, but it is not clear whether POL was used in any prosthesis usage analyses.

6. In the results section titled "Prosthesis representation doesn't depend on the users' prosthesis type", other key factors of prosthesis use were not considered when analyzing prosthesis categorization: level of injury (transradial vs transhumeral), duration of time since limb loss, congenital vs acquired, and duration of time using primary prosthesis. Given the potential role of visual learning on the results, the duration of time since starting to use their primary prosthesis may be especially important to investigate in the future.

7. Page 10 - "A group comparison between users of the two prosthesis types revealed no significant effect." It is not clear what metric is being compared between users of cosmetic and active prostheses in this section.

---

## [Decision Letter · Decision Letter 3]

7 Apr 2020

Dear Dr Makin,

Thank you for submitting your revised Short Report entitled "Is an artificial limb embodied as a hand? Brain decoding in prosthetic limb users" for publication in PLOS Biology. I have now obtained advice from original reviewers 1 and 4 and have discussed their comments with the Academic Editor. You will note that reviewer 1, Surjo Soekadar, has signed his comments.

I'm delighted to let you know that we're now editorially satisfied with your manuscript. However, before we can formally accept your paper and consider it "in press", we also need to ensure that your article conforms to our guidelines. A member of our team will be in touch shortly with a set of requests. As we can't proceed until these requirements are met, your swift response will help prevent delays to publication. Please also make sure to address the data and other policy-related requests noted at the end of this email.

*Copyediting*

*Published Peer Review History*

*Early Version*

*Submitting Your Revision*

Sincerely,

Gabriel Gasque, Ph.D., 

Senior Editor

PLOS Biology

ETHICS STATEMENT:

-- Please indicate if the informed consent was written. If not, please explain why.

DATA POLICY:

Note that we do not require all raw data. Rather, we ask for all individual quantitative observations that underlie the data summarized in the figures and results of your paper. For an example see here: http://www.plosbiology.org/article/info%3Adoi%2F10.1371%2Fjournal.pbio.1001908#s5

These data can be made available in one of the following forms:

Regardless of the method selected, please ensure that you provide the individual numerical values that underlie the summary data displayed in the following figure: Figure 2A, 3ABCEF, S1, S2BCD, S3AB, and S5AB. 

Please also ensure that figure legends in your manuscript include information on where the underlying data can be found and ensure your supplemental data file/s has a legend.

Reviewer remarks:

Reviewer #1, Surjo R. Soekadar: Makin et al. present the third revision of their manuscript on prosthesis embodiment analyzed in terms of multidimensional fMRI activation patterns in response to visual depictions of prostheses, hands, and tools. The remaining concerns of all reviewers have been sufficiently addressed. The core outcome measure presented by the authors consisted of a complex distance metric measuring the degree to which prosthesis representations moved away from those of hands and tools, and towards each other (a distinct new category) in prothesis users. Many comments requested that the pairwise distances making up this complex distance metric should be further explained, which the authors have now satisfactorily done. The manuscript is therefore now suitable for publication. 

Reviewer #4: I thank the authors for thoroughly addressing all of my comments. I appreciate the addition of Supplementary Table 4 showing the results of a PCA examining the contributions of pairwise-distances on the prosthesis-similarity index. I have no further concerns and recommend the manuscript for publication.

---

## [Editor Report · Decision Letter 4]

20 May 2020

Dear Dr Makin,

On behalf of my colleagues and the Academic Editor, Karunesh Ganguly, I am pleased to inform you that we will be delighted to publish your Short Reports in PLOS Biology. 

Early Version

PRESS 

Kind regards,

Vita Usova

Publication Assistant, 

PLOS Biology

on behalf of

Gabriel Gasque,

Senior Editor

PLOS Biology